

# Effectiveness of low impact development for urban inundation risk mitigation under different scenarios: a case study in Shenzhen, China

Jiansheng Wu[1,2], Rui Yang[1], Jing Song[3]

[1]Key Laboratory for Urban Habitat Environmental Science and Technology, Shenzhen Graduate School, Peking University, Shenzhen 518055, PR China
[2]Key Laboratory for Earth Surface Processes, Ministry of Education, College of Urban and Environmental Sciences, Peking University, Beijing 100871, PR China
[3]Department of Urban Planning and Design, The University of Hong Kong, Hong Kong

*Correspondence to*: Jing Song (songjing@hku.hk)

**Abstract.** The increase in impervious surfaces associated with rapid urbanization is one of the main causes of urban inundation. In order to eliminate the adverse effects caused by impervious surfaces, many scholars have begun to research the use of low impact development (LID) practices to mitigate urban inundation risk. This study proposes a hydrodynamic

inundation model, coupling SWMM (Storm Water Management Model, 1D) and IFMS Urban (Integrated Urban Flood Modelling System, 2D), to simulate inundation depth, area, and time of stormwater inundation on an urban watershed scale, as well as to assess the effectiveness of two LID practices, permeable pavement (PP) and green roof (GR), under 25 % GR + 25 % PP, 50 % GR + 50 % PP, 75 % GR + 75 % PP, 100 % GR + 100 % PP, 100 % PP, 100 % GR scenarios, and Low, Medium, High hazard levels. The results show the following. 1) LID practices can effectively eliminate inundation risk for

most areas under Low hazard level for urban inundation. They can ease the inundation risk for places under higher hazard levels for urban inundation under different scenarios. More specifically, the maximum inundation depth was reduced by 14-29 %, inundation areas were reduced by 34-55 %, and average inundation time was reduced by 0-43 % in six scenarios. 2) In this study, the performance of PP is better than that of GR under different scenarios and hazard levels. 3) The scenario of 100 % PP + 100 % GR has the best effectiveness for inundation reduction, but that of 25 % PP + 25 % GR is more efficient

when considering cost-effectiveness. The results of this study can serve as a reference to local governments, and provide suggestions regarding urban inundation control, disaster reduction, urban renewal, and so on.

## 1 Introduction

In recent years, urban stormwater inundation hazards have occurred frequently in a number of major cities all over the world, leading to huge losses in local areas (Bhattarai et al., 2016). In China, according to a report by the Ministry of Housing and

30 Urban-Rrural Development (MOHURD) in 2010, 62 % of 351 cities have suffered from inundation hazards, and 137 of them



faced the negative effects of urban floods on more than three occasions from 2008 to 2010. In 2012, 2013, 2014, and 2015, the number of cities that suffered urban inundation was 184, 234, 125, and 154, respectively, including Beijing, Shanghai, Guangzhou, Shenzhen, and other large cities. Urban inundation and secondary disasters associated with it are increasingly threatening to the sustainable development of urban areas.

Some researches point out that, in addition to extreme precipitation and low standards for urban drainage infrastructure, rapid urbanization has become an important cause of frequent urban stormwater inundation (Arnold, 1996; Beckers et al., 2013;Claessens et al., 2006; Zahmatkesh et al., 2015b). The rapid expansion of the city generally leads to an increase in impervious surfaces, which makes the hydrological characteristics of the urban surface change a lot (Arnold, 1996; Jacobson, 2011; Rose and Peters, 2001). On the one hand, impervious surfaces replaced rivers, lakes, green spaces, and urban forests,

as they weaken the flood control capability of urban system for infiltration, evaporation, filtration, and storage (Hao et al., 2015; Jacobson, 2011; Meyer, 2001); on the other hand, the expansion of impervious areas accelerates rainwater convergence on urban surfaces, resulting in increased runoff and peak flow (Hatt et al., 2004; Leopold et al., 1995; Liu et al., 2015). The increase in runoff and peak flow put pressure on urban drainage facilities and exacerbate the risk of urban inundation.

To solve the problem of urban inundation, scholars put forward the "Sponge City" initiative, which allows cities to act as a sponge to filtrate, purify, evaporate, and store rainwater (Mao et al., 2017; Sang and Yang, 2016). As one of the important development concepts, low impact development (LID) has been applied in sponge city construction (Luan et al., 2017) and is widely applicable to reducing the impacts of urban inundation associated with rapid urbanization (Dietz and Clausen, 2008; Dietz, 2007; Xia et al., 2017; Zahmatkesh et al., 2015a). LID is a stormwater management strategy that uses microscale and

localized practices to control the runoff and pollution caused by a storm (Damodaram et al., 2010; EPA, 2000; HUD, 2003). Since the 1990s, LID practices have been widely used in Europe, the United States, and some other developed countries, and the types of LID practices have been enriched to include permeable pavements, green roofs, bioretention, swales, infiltration wells/trenches, infiltrating wetlands, and rain barrels (Hunt et al., 2010).

The hydrological effectiveness of LID practices has been further researched through field and laboratory studies (Abbot and

Comino-Mateos, 2003; Berndtsson, 2010; Davis, 2008; Davis et al., 2012; Fassman and Blackbourn, 2010). For example, Hood et al. (2007) monitored low impact residential development and traditional residential development in the town of Waterford, Connecticut, USA, and found that LID practices helped lower runoff, peak flow, and discharge volume. Dreelin et al. (2006) designed a test to compare the performance of asphalt and permeable pavement parking lots in Athens, Georgia, USA, and the results showed that the porous lot contributed 93 % less runoff than the asphalt lot during natural storm events.

Bliss et al. (2009) constructed and monitored a green roof in Pittsburgh, Pennsylvania, USA, and reported that runoff was reduced by up to 70 %, peak flow reduced by 5-70 %, and the hydrograph was delayed by several hours when comparing a green roof to a normal roof for the same building.

Considering the value of exploring the effectiveness of LID practices in actual situations, many scholars have focused on simulations at a larger scale, such as watersheds (Ahiablame et al., 2012; Dietz and Clausen, 2008; Roy et al., 2008;



Salvadore et al., 2015). For example, Palla and Gnecco (2015) reported that the LID combinations of green roof (GR) and permeable pavement (PP) could decrease runoff and peak flow by 23 % and 45 %, respectively, and delay the hydrograph by up to 19 % at the urban catchment scale. Trinh and Chui (2013) conducted a simulation and found that GR could reduce the peak flow by 50 % and delay the hydrograph by 2 h, bio-retention (BR) systems could reduce the peak flow by 50 %, and the

5 effectiveness of combinations of GR and BR systems even could reduce the peak flow to the pre-urbanized level. Morsy et al. (2016) reported that rain gardens could mitigate runoff to approximately 15, 27 , and 38 % for 2-, 5-, and 10-year storm events, respectively, which reduced the flood risk in watersheds. Ahiablame et al. (2013) assessed the effectiveness of rain barrel/cistern and porous pavement in two urbanized watersheds near Indianapolis, Indiana, USA, and through simulations, they find that LID practices reduced runoff and pollutant loads; they list some scenarios of LID combinations that are good

retrofitting options for local areas.

It is noteworthy that peak flow reduction, runoff reduction, and hydrograph delay are widely used indexes when evaluating the performance of LID practices for mitigating urban inundation risk (Ahiablame and Shakya, 2016; Qin et al., 2013; Zhang et al., 2016). However, these indexes are not very intuitive; in fact, the spatial distribution of urban inundation and its changes with rain time are more beneficial to local residents, such as providing guides for their travel behaviours. Some

15 recent studies have constructed 2D models to simulate the spatial distribution of surface inundation and evaluate the risks of inundation (Hu et al., 2017; Wu et al., 2017). However, we find that few researchs use hydrodynamic models, like SWMM, which not only can realize the simulation of the spatial distribution of urban inundation but also can explore the dynamic effectiveness of LID practices on inundation mitigation. Further, existing literature seldom explores the efficiency of LID practices under different scenarios (LID combinations), which can provide support for LID practice construction for areas

vulnerable to urban inundation. Therefore, we aim to fill these gaps by conducting this study. In order to explore the performance of LID practices for mitigating impacts of urban inundation, we establish a 2D hydrodynamic model to evaluate the inundation depth, area, and time of PP and GR, two widely used LID practices, under different scenarios and hazard levels, and evaluate the efficiency of every scenario. This study enriches the inundation mitigation research of LID on an urban watershed scale and provides some references to urban stormwater management and inundation mitigation.

## 25  2 Materials and methodology

### 2.1 Study site

Shenzhen is located in the coastal area of Guangdong Province in southern China (Fig. 1). Belonging to the subtropical maritime monsoon climate, Shenzhen is hot and rainy in summer and mild in winter, and the average annual rainfall is 1837 mm. April to September marks the rainy season in Shenzhen, and during this period, precipitation is concentrated and

30 stormwater overflow is frequent. Accordingly, urban inundation is particularly serious in this period, which leads to inconvenient and economic losses to local residents, and even the loss of lives.



The study site is located in Guangming New District of Shenzhen, China (Fig. 1). Because of the heavy inundation disasters, Guangming New District was identified as the first pilot area for LID practices in Shenzhen in October 2011 by MOHURD. To date, 17 LID practices have been completed in Guangming New District. Thus, this provides us the opportunity to check the effectiveness of LID practices on urban inundation mitigation.

The study site is a rapid urbanization zone of Guangming New District, about 37.68 km$^2$, and located in the Maozhou River Basin. At this study site, construction land area is 26.31 km$^2$, which accounts for 69.8 % of the total area. Using the investigation and land use map shown in Fig. 1, we find that the developed areas are dominated by industrial land and residential land, and this intensive development easily led to urban inundation during the heavy rainy season, such as during the heavy rain on 11 May 2014, with 144.9 mm of rainfall within 24 h and a maximum hourly rainfall of 23.6 mm, which
caused serious urban inundation and great loss to the residents and production.

## 2.2 Data

The input data needed for modelling mainly include inundation, land use, digital elevation model (DEM), weather, and pipe network data.

1. The land use data (2013 year) and pipe network data are provided by the Shenzhen government. According to remote
sensing images and the needs of model building, we make a generalization for the original data and divide the study area into water, low density construction land, high density construction land, bare land, woodland, grassland, and agricultural land, namely, seven land use types in total (Fig. 1).

2. The DEM of study area (Fig. 2) is downloaded from Geospatial Data Cloud, and the resolution is 30 m. In order to correspond to the size of the grid in IFMS Urban model, we resampled the DEM to 15 m×15 m in ArcGIS.
3. The weather data are sourced from the Shenzhen Meteorological Data System (https://data.szmb.gov.cn/). We chose two rainstorm event datasets, from 11 May 2014 and 10 May 2016 (Fig. 3) for model simulation, which included the volume of rainfall every hour. The inundation data were obtained from Shenzhen SanFang (flood, drought, and wind defence) headquarters and the Guangming New District Urban Construction Bureau.

4. Because the urban pipe network is intricate, we deleted some redundant and incorrect data and retained the major nodes
and pipelines. Finally, the 4502 pipelines and 1175 nodes in this study site were generalized to 597 pipelines and 653 nodes, respectively, including 56 outlets and 597 inspection nodes (Fig. 2).

## 2.3 SWMM model

This study uses SWMM (Storm Water Management Model, 1D) to construct a 1D sewer model. SWMM, based on hydrology and hydrodynamics, is an urban storm water management model developed by the US EPA (Environmental
Protection Agency). SWMM can simulate the dynamic runoff quantity and quality from primarily urban areas, and it has been widely used to simulate the hydrologic performance of specific types of LID practices (Rossman, 2010; Wu et al., 2013).





The building processes of the SWMM model are shown in Fig. 4. Some measurement parameters, such as area, slope, impermeability, etc., of sub-catchments can be calculated with formulas. Other parameters, such as the Manning coefficient, depression store, etc., must be calibrated several times to be determined. First, referring to the SWMM Model Manual and other literature (Rossman, 2010; Wu et al., 2017), we determine the reference range of these parameters. Then, according to

the reported urban inundation data, we calibrate the model to obtain relatively accurate parameters.

## 2.4 Coupling the SWMM/IFMS Urban models

SWMM can simulate the dynamic rainfall runoff process, but it cannot simulate the spatial distribution (2D) of surface inundation. Recently, some scholars have conducted some experiments using secondary development on this software (Seyoum et al., 2012; Son et al., 2016; Zhu et al., 2016), but due to differences in computer programming, it might be

difficult to copy these applications to other urban areas. Therefore, coupling a model with SWMM and the other models that can realize the 2D simulation is another way to simulate the spatial distribution of urban inundation (Huong and Pathirana, 2013; Wu et al., 2017).

This study innovatively selects SWMM and IFMS Urban (Integrated Urban Flood Modeling System, 2D) to carry out a 2D simulation model. IFMS Urban was developed by China Institute of Water Resources and Hydropower Research (IWHR) in

cooperation with other institutions. Through meshing the study area into grids, IFMS Urban can analyse urban inundation, and it has great compatibility with ArcGIS and SWMM. IFMS Urban considers the 2D shallow water equations during its calculation process for urban inundation simulation, and it also considers the coupling effect between the urban pipe network and the grids when simulating urban inundation. Therefore, this system can be applied for urban inundation simulation. This study innovatively to couple SWMM and IFMS Urban to simulate the spatial distribution of inundation and to explore the

effectiveness of LID practices.

First, we divide the study area into quadrilateral grids. As the smallest calculation unit, the grid's edge is approximate 15 m. Then we assign elevation to the grids. Regrettably, we do not have a high precision DEM for the entire research area; therefore, we cannot assign elevations for the areas that are easily inundated. After the field investigation, however, we found that most inundation areas are on the streets, and we have accurate elevation data for the manholes (nodes) on the

ground. Using the elevation date of these nodes, we obtain the ground elevation of the streets in our research area through Kriging interpolation with the help of ArcGIS. And then we assign the elevation of the streets to the grids. Finally, we build a 2D inundation analysis model coupled SWMM and IFMS Urban (Fig. 4).

## 2.5 Scenarios of LID combinations for simulation

Considering the feasibility and representativeness of LID practices for urban inundation mitigation, this study chose two

types of LID practices, GR and PP, to simulate and explore their effectiveness on urban inundation mitigation. Through remote sensing images and field investigations, we found that it is impracticable to add LID practices to the surfaces of high density construction lands given their development strength. Therefore, this research sets principles for the construction of



LID practices: GR can only be built on low density construction lands, and PP can be built both on low and high construction lands as well as on some streets. According to these principles, we set a series of proportions from 25 % to 100 % representing the construction strength of different types of LID combinations to simulate and explore the effectiveness of LID practices for mitigating urban inundation under different scenarios. Finally, a benchmark and six scenarios were designed, and the parameters for LID practices (Chui et al., 2016; Cipolla et al., 2016; Qin et al., 2013; Zhang, 2015) are shown in Table 1:

Benchmark: No LID practices

Scenario 1: 25 % GR + 25 % PP

Scenario 2: 50 % GR + 50 % PP

Scenario 3: 75 % GR + 75 % PP

Scenario 4: 100 % GR + 100 % PP

Scenario 5: 100 % PP

Scenario 6: 100 % GR

## 3 Results

### 3.1 Model calibration and validation

The coupled model is calibrated using the rainfall-inundation data from 11 May 2014. Based on the relevant literature and the SWMM manual, we determined the final SWMM parameters (Table 2) through several calibration iterations. From the final calibration results (Table 3), we can find that except for Inundation site Gm 20, the absolute value of the maximum inundation depth between the observed and simulated value is in the range of 0-0.14 m, and the relative error is 0-30 %.

The rainfall-inundation data on 10 May 2016 was chosen to further validate the coupled model. Based on the actual urban inundation data on that day from the Guangming New District Urban Construction Bureau, there are three valid datasets to be simulated with the coupled model. From Table 3, the results show that the absolute value of the difference in maximum inundation depth between the observed and simulated is 0.04 m (Gm 11), 0.05 m (Gm 12) and 0.02 m (Gm 20), and the relative errors are 20, 7, and 5 %, respectively. According to similar research (Wu et al., 2017), the calibration and validation results of the model are acceptable for simulating rainfall-inundation.

### 3.2 Inundation depth under different scenarios

Figure 5 and Table 4 show the simulation results of inundation depths under different scenarios. Compared to the benchmark, the reduction rates of maximum inundation depth are 16, 22, 26, and 29 %, respectively, when the proportion of LID combinations increases from 25 to 100 %. And the results between 100 % PP and 100 % GR show that PP and GR almost have the same performance at the maximum inundation depth, and both of the reduction rates are 14 %.



To further explore the impacts of LID practices on inundation mitigation, we set three hazard levels in terms of the depths of urban inundation: Low (< 0.2 m), Medium (0.2 m-0.4 m), and High (>= 0.4 m), according to the literature (Su et al., 2016) as well as actual situation of the study area. Compared to the benchmark, the range of average inundation reduction depths at Low, Medium and High levels were 0.04-0.06 m, 0.07-0.14 m, and 0.11-0.19 m, respectively under Scenarios 1 to 4.

Correspondingly, the range of average inundation reduction rates were 60~80 %, 27~54 %, and 22~40 % at Low, Medium and High levels under Scenarios 1 to 4 (Fig 6). Based on the simulation results of these four scenarios, we clearly note that under different hazard levels, the average inundation reductions and reduction rates increase as the proportion of LID combinations increases. Additionally, we clearly see that reductions for the Low level are 0.07, 0.10, 0.12, and 0.13 m lower than those of the High level under Scenarios 1 to 4, respectively, while the reduction rates at the Low level are 38, 44, 43,

and 40 % higher than those at the High level under Scenarios 1 to 4, respectively. This means that the reduction effects become more evident as hazard level increases, while reduction rates decrease. This is due to the fact that few reductions in Low risk areas will result in improvements, which means the roles of LID practices with respect to urban inundation mitigation are less obvious at High levels than those at Low levels.

Further, we determine that the performances of 100 % PP and 100 % GR are between scenarios of 25 % GR + 25 % PP and

50 % GR + 50 % PP under different hazard levels. These results suggest that the effectiveness of LID combinations might be better than that of a single type of LID practice. Based on the comparison of the two LID practices, we find that the reduction rates of 100 % PP are 6, 7, and 2 % higher than those of 100 % GR at Low, Medium, and High levels, respectively. This means that PP might perform better than GR in inundation depth reduction.

### 3.3 Inundation areas under different scenarios

Figure 6 shows inundation area changes under different scenarios and hazard levels. According to the simulation results of Scenarios 1 to 4, the range of inundation areas are 116-79.1, 8.7-4.9, and 0.6-0.2 ha under Low, Medium, and High levels, respectively. Compared to the benchmark (167.6, 19.5, and 2.1 ha), the range of reduction rates are 31-53, 55-75, and 71-90 % for Low, Medium, and High levels under Scenarios 1 to 4, respectively. It is clear that the impacts of inundation are reduced under different hazard levels since adding LID practices onto the original land use. The reduction rates under the High level

which are up to 71-90 %, seem to be more obvious. This means that although inundation areas under the High level have the lowest reduction rates for inundation depth, most of them can be effectively reduced to lower hazard levels (Medium or Low).

For 100 % PP and 100 % GR, the reductions of inundation areas are similar to 25 % PP + 25 % GR, which further proves that LID combinations are more effective than single LID practices. At Low and Medium levels, the inundation areas of 100 %

PP are 9 ha and 1.6 ha less than those of 100% GR, while both of them perform the same at the High level, which means that despite inundation depth or inundation area, both of PP and GR perform the same in High level areas.




## 3.4 Inundation time under different scenarios

Inundation time represents inundation risk from another perspective. From Scenarios 1 to 4, we clearly see that the inundation time under Medium and High levels is longer than that under the Low level given the same scenario (Table 5), which reflects increased danger under Medium and High levels than under a Low level. As the proportion of LID combinations increases, the average inundation time decreases under the three hazard levels. 100 % PP and 100 % GR perform a little better than Scenario 1, and the inundation time of 100 % PP is 1.3 h less than 100 % GR.

It is worth noting that, compared to the benchmark, the inundation time of Low and Medium levels in Scenario 1 increases slightly, while it only decreases a little at the High level. This phenomenon does not mean that the performance of LID practices is not useful for decreasing inundation time or that the model has errors. From the above-mentioned analyses of inundation depth, area, and time, we can know that areas under low risk to urban inundation are easily improved. It is undeniable, however, that there are still some inundation areas having a long inundation time that are difficult to mitigate. From Fig. 5 we can see that most of them are located in areas where the drainage infrastructures are not perfect and LID practices are not arranged. Thus, because of the mitigations of many short-time inundation areas, the average inundation time rises from 4 to 4.1 h in this scenario. As the proportion of LID combinations increases, the inundation areas are mitigated, and the inundation time decreases from 4.1 to 2.3 h for 100 % PP + 100 % GR, in total.

## 4 Discussion

### 4.1 Performance of PP and GR

To ensure the effectiveness of LID practices for urban inundation mitigation is very important for stormwater management. From the above-mentioned analysis, we find that the effectiveness of PP for urban inundation mitigation performs better than that of GR in terms of the three indexes. Many studies have also proven that PP has better performance than GR in runoff reduction (Zhang et al., 2016) and urban flooding mitigation (Qin et al., 2013). Objectively speaking, except for the effectiveness of LID parameters, the size, spatial pattern, and other factors may also have an impact on the performance of LID practices. Therefore, the performances of PP and GR are different for different study areas. The findings of this study suggest some advantages of PP that might suit local developed areas very well, which can provide some suggestions to local stormwater management officials. The findings also prove that compared to a single LID practice, combinations of LID practices should be applied at the local community level for urban inundation mitigation.

### 4.2 Efficiency of LID practices

Through scenario simulations, the performance of LID practices for urban inundation mitigation on the urban watershed scale has been explored comprehensively. We found that 25 % PP + 25 % GR was the best choice for inundation mitigation in these scenarios for the selected research area, though its performance was not the best. Compered to benchmark, 25 % PP





+ 25 % GR reduced the maximum inundation depth by 14 % and the total inundation areas by 34 %, while 100 % PP + 100 % GR reduces the maximum inundation depth by 29 % and the total inundation areas by 55 %. It's clear that the efficiency of 25 % PP + 25 % GR is higher than that of other scenarios. Therefore, when considering cost-effectiveness on inundation mitigation, the best LID combination is about 25 % in this study area.

This study also found a limitation for the application of LID practices. For example, in the Low risk areas, when the percentage of PP and GR increases from 25 % to 50, 75, and 100 %, the average inundation reduction rate rises from 60 % to 74, 79, and 80 %, respectively. It is clear that the reduction rate grows slowly while the percentage increases proportionally, which means the marginal benefits of LID decrease. The same phenomenon also occurs in Medium and High risk inundation areas.

The phenomena described above indicate that the risk in some inundation areas is difficult to mitigate in the study area, especially in places with low terrain or poor infrastructure. For these areas, the continuous increase of the construction strength of LID practices evidently cannot mitigate the risk of urban inundation; instead, it will decrease the efficiency of LID practices in the whole urban watershed.

## 4.3 Limitations and future studies

Our study site is large but lacks accurate data for depth of urban inundation, which limits the accuracy of parameter calibration and validation, and further limits the accuracy of the simulation results. Furthermore, the simulation is simplified without considering the roles played by pumping stations and the river networks for urban inundation mitigation. Although most existing research has similar problems (Hu et al., 2017; Wu et al., 2017), we still think the accuracy of the simulation needs to be improved for future studies.

In China, urban inundation seems to be more and more serious, and LID practices should be focused on as efficient strategies for urban inundation mitigation. At present, most research focuses on a number of LID practices that play a dominant role on urban inundation mitigation. However, we also find that the spatial distribution of landscape patterns also contributes to urban flooding mitigation (Kim and Park, 2016; Giacomoni and Joseph, 2017). This provides a new perspective for further research on the effectiveness of LID practices on urban inundation mitigation. In addition, determining how to effectively

integrate LID practices into urban development (Chui et al., 2016), especially for places extremely vulnerable to urban flooding, is still worth discussing in the future.

## 5 Conclusion

We constructed a 2D inundation model that coupled SWMM and IFMS Urban on the urban watershed scald and evaluates the effectiveness of LID practices for mitigating urban inundation under different scenarios and hazard levels. The

conclusions are described below: First, LID practices can effectively eliminate most inundation risk at the Low level and ease the inundation risk at higher levels under different scenarios. Compared to the benchmark, the simulation results




suggest that the maximum inundation depth can be reduced by 14-29 %, the total inundation area can be reduced by 34-56 %, and the average inundation time can be reduced by 0-43 %. Second, the mitigation effectiveness of 100 % PP is better than that of 100 % GR in terms of inundation depth, inundation area, and inundation time under different scenarios and hazard levels. Further, PP is suitable for application to reduce the impacts of urban inundation for local areas. Third, combinations

of LID practices are more effective for mitigating urban inundation than single LID practices. The effectiveness of inundation reduction under the scenario of 100 % PP + 100% GR is the best among the six scenarios; however, its efficiency is the lowest. In the contrast, 25 % PP + 25 % GR has good performance when considering the effectiveness for mitigating inundation and the construction of LID practices, which means the best LID combination is about 25 % in this study area. Facing urban inundation comprehensively using a variety of stormwater management measures may be the most effective

method.

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





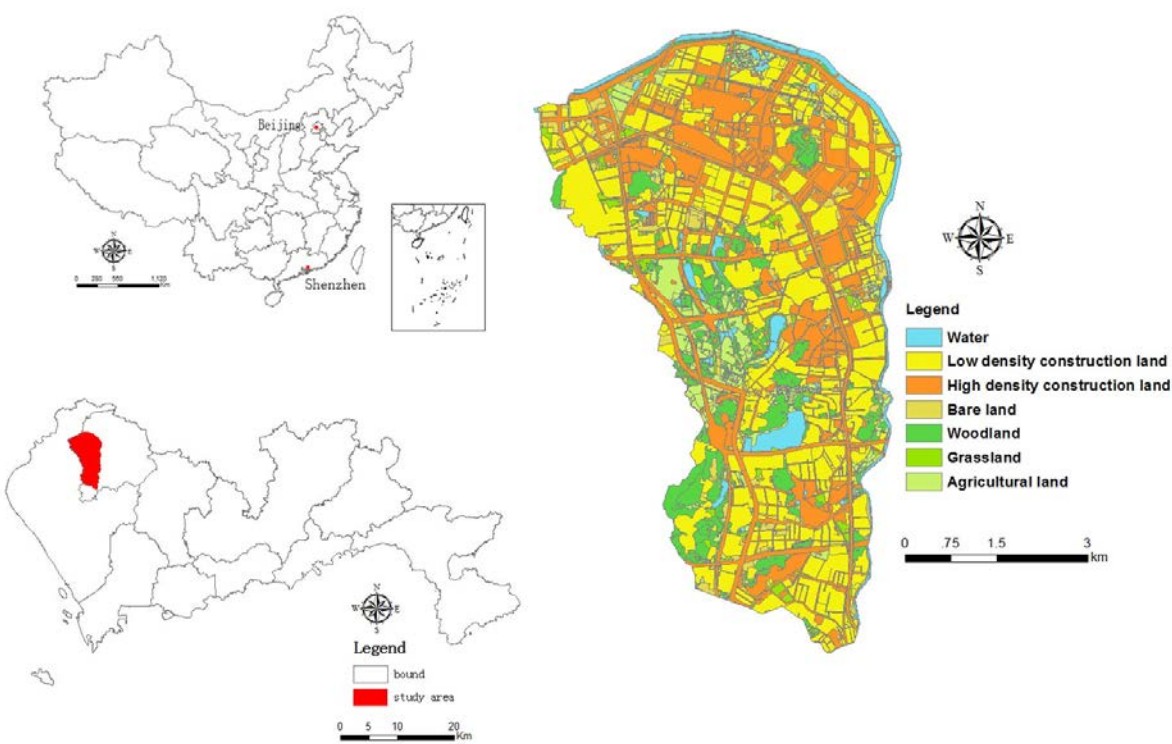

**Figure 1: Location and land use map of the study area in Guangming New District of Shenzhen, China.**

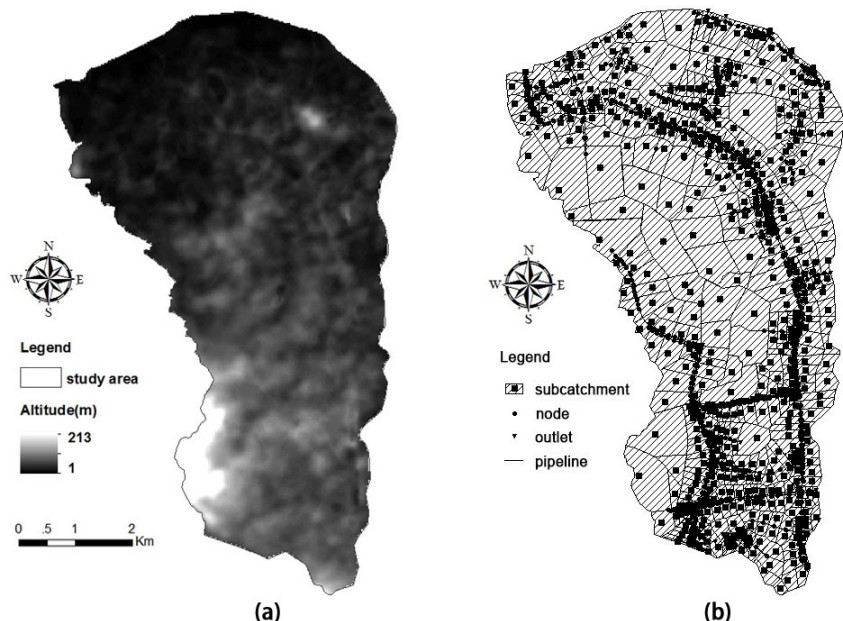

**Figure 2: Altitude (a) and SWMM model (b) of the study area.**





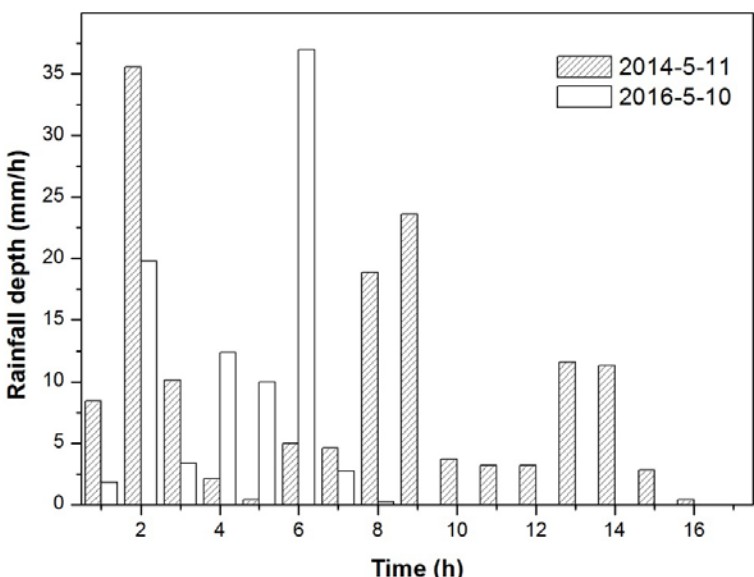

**Figure. 3: Rainfall intensity for the events on 11 May 2014 and 10 May 2016 in the study area.**

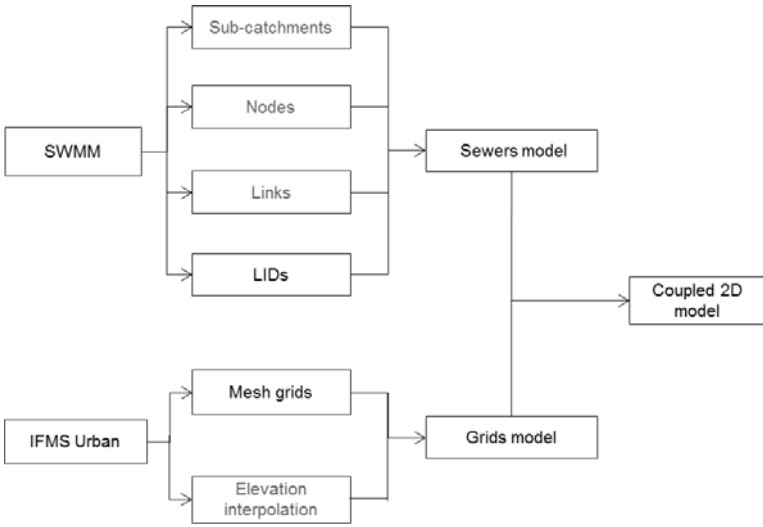

**Figure. 4: Processes of coupled inundation model building.**



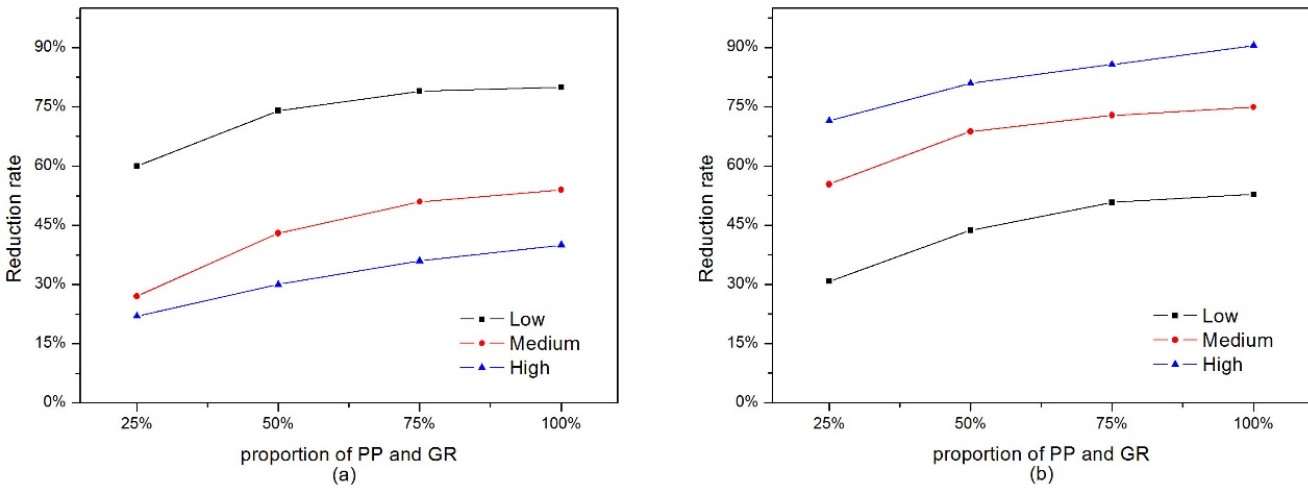

**Figure. 5: Inundation depth maps of the study area under different scenarios.**

**Figure. 6: Reduction rates of average inundation depth (a) and inundation areas (b) under different scenarios and hazard levels.**




**Table 1 LID parameters in SWMM.**

| LID types | structure | parameter | value |
|---|---|---|---|
| PP | Surface | Berm height (mm) | 2 |
| | | Vegetation volume fraction | 0 |
| | | Surface roughness (Manning's n) | 0.014 |
| | | Surface slope (%) | 1 |
| | Pavement | Thickness (mm) | 100 |
| | | Void ratio (Voids/Solids) | 0.25 |
| | | Impervious surface fraction | 0 |
| | | Permeability (mm/h) | 250 |
| | | Clogging factor | 0 |
| | Storage | Thickness (mm) | 150 |
| | | Void ratio (Voids/Solids) | 0.4 |
| | | Seepage fate (mm/h) | 1.2 |
| | | Clogging factor | 0 |
| GR | Surface | Berm height (mm) | 3 |
| | | Vegetation volume fraction | 0.1 |
| | | Surface roughness (Manning's n) | 0.017 |
| | | Surface slope (%) | 1 |
| | Soil | Thickness (mm) | 100 |
| | | Porosity (volume fraction) | 0.5 |
| | | Field capacity (volume fraction) | 0.2 |
| | | Wilting point (volume fraction) | 0.024 |
| | | Conductivity (mm/h) | 30 |
| | | Conductivity slope | 5 |
| | | Suction head (mm) | 60 |
| | Drainage mat | Thickness (mm) | 3 |
| | | Void fraction | 0.5 |
| | | Roughness (Manning's n) | 0.1 |

**Table 2 Primary calibrated parameters in SWMM.**

| SWMM parameters | calibrated value |
|---|---|
| N-Imperv | 0.015 |
| N-Perv | 0.15 |
| Dstore-Imperv/mm | 2 |
| Dstore-Perv/mm | 5 |



| Zero-Imperv/% | 25 |
|---|---|
| Roughness | 0.013 |
| Max.Infil.Rate/(mm/h) | 76 |
| Min.Infil.Rate/(mm/h) | 12 |
| Decay Constant | 2 |
| Drying Time | 5 |

**Table 3 Comparison of inundation depth between the observed and simulated results.**

| Inundation site | Storm on 11 May 2014 | | | Storm on 10 May 2016 | | |
|---|---|---|---|---|---|---|
| | Reported | Simulated | RE (%) | Reported | Simulated | RE (%) |
| Gm 11 | 0.25 | 0.32 | 28 | 0.2 | 0.24 | 20 |
| Gm 12 | 0.55 | 0.69 | 25 | 0.7 | 0.75 | 7 |
| Gm 20 | 0.5 | 0.24 | -52 | 0.4 | 0.42 | 5 |
| Gm 21 | 0.45 | 0.46 | 2 | - | - | - |
| Gm 24 | 0.2 | 0.26 | 30 | - | - | - |
| Gm 22 | 0.2 | 0.2 | 0 | - | - | - |
| Gm 16 | 0.2 | 0.23 | 15 | - | - | - |

"-" means data miss, "RE" means "relative error", unit: m.

**Table 4 Maximum inundation depth under different scenarios.**

| | Bench mark | 100 % PP | 100 % GR | 25 % PP+25 % GR | 50 % PP+50 % GR | 75 % PP+75 % GR | 100 % PP+100 % GR |
|---|---|---|---|---|---|---|---|
| maximum inundation depth (m) | 0.69 | 0.59 | 0.59 | 0.58 | 0.54 | 0.51 | 0.49 |
| Reduction rate (%) | - | 14 | 14 | 16 | 22 | 26 | 29 |

**Table 5 Inundation time under different scenarios and hazard levels.**

| | benchmark | 100 % PP | 100 % GR | 25 % PP+25 % GR | 50 % PP+50 % GR | 75 % PP+75 % GR | 100 % PP+100 % GR |
|---|---|---|---|---|---|---|---|
| Low (h) | 3.4 | 3.3 | 3.3 | 3.7 | 3.3 | 2.5 | 2.2 |
| Medium (h) | 7.7 | 7.5 | 7.7 | 8.2 | 7.1 | 6 | 4.7 |
| High (h) | 10.6 | 9.3 | 8.4 | 9.6 | 7.6 | 6 | 4.7 |
| Total (h) | 4 | 3.6 | 3.6 | 4.1 | 3.6 | 2.8 | 2.3 |