# Peer review of "Effectiveness of low impact development for urban inundation risk mitigation under different scenarios: a case study in Shenzhen, China"

_Natural Hazards and Earth System Sciences, 2017_

## Referee Comment (RC1) · Anonymous Referee #1 · 2 Feb 2018

General comments This study sought to evaluate the impacts of LID practices on urban inundation at a watershed scale in China. Extensive modeling was used to assessed various LID implementation scenarios with a hydrodynamic inundation model, which coupled SWMM and IFMS Urban models. The study is interesting and will contribute to the understanding of LID effectiveness related to flood reduction. However, the scientific quality and presentation quality were poor. First, English in this paper is poor. some contents are difficult to understand. I would strongly recommend the editing by an experienced or even better native English speaker. Next there some major and obvious weakness in methodology and results. I listed them below. Also, it requires lots of improvements in other sections.

[Figure]

Introduction: Review should be correct. In page 3 line 16, "we find that few researches use hydrodynamic models, like SWMM ...". In fact, there are many studies of SWMM in LID field, especially in 2017 in China. Also, the introduction is very universal, does not clearly lead to the specific content of the manuscript and is missing a central theme. For readers to quickly catch your contribution, it would be better to highlight major difficulties and challenges, and your original achievements to overcome them in a clearer way.

Materials and methodology:

1) Why you selected these two events? Were they have special characteristicsïij§

2) How you downscaled the dem resolution? The bias from downscaling was corrected?

3) land use area should be described as well as the implementation area of each LID scenario

3) Is there discharge Data for SWMM calibration?

4) Why you coupled SWMM and IFMS Urban models? What the advantages compared with others? This study discussed inundation depth, area and time. There three indices could be got from some 2D inundation model. As I know, the outputs of SWMM are outflow, peak flow, flood volume, etc. This study didn't mention any of them. So, why you need SWMM?

Results:

1) The results for hazard level seem very sensitive to the thresholds chosen. Please give information on the thresholds chosen.

2) Results are contradicted. The authors reported on Page 7 line 11-12 "the reduction effects become more evident as hazard level increases", "the roles of LID practices with respect to urban inundation mitigation are less obvious at High levels than those

at Low levels". So which one is correct?

3) please show the spatial distribution of reductions in inundation depths instead of average reduction

4) please give more information of PP and GR implementation area, otherwise, you cannot say PP performs better than GR

5) in section 3.3 you said the reduction in inundation area under High level was more obvious, but in section 3.1, reduction in inundation depth was less obvious. Please explain.

6) please show the spatial distribution of reductions in inundation time instead of average

7) one of the key points in your study is to compare the differences of all scenarios at three hazard levels not to find the differences among three hazard levels.

Discussion: 1) The discussion is lacking depths. What are the same and different points comparing your study and others? What you studied from this research.

2) The discussion on cost-effectiveness completely fell from the sky on page 9 line 3. You neither present how the costs were estimated nor discussed them in the Results.

3) In page 9 line 22, "we also find that spatial distribution of landscape patterns ...". This information completely fell from the sky. You neither present them in the Results.

4) You reported 25% of PP and GR had the highest efficiency. Is it correct? Do you consider the effect of rainfall intensity and frequency? LID effectiveness is highly related to rainfall intensity and frequency.

Specific comments

1) Page (P) 1 line (L) 15-19, too long to understand

2) P1L25, considering cost-effectiveness, you don't give any information on it.

3) P2L4: what are secondary disasters? it is better to delete

4) P3L18-20, there some studies on this topic, please review them

5) P3L29, give rainfall information from April to September

6) P4L5-10. simplify the description

7) P4L12, delete"needed for modeling"

8) P4L18-19, how to do

9) P4L20-22, improve

10) P5L1-3, improve

11) give clear information on the model

12) P5L31-32, "we found ..."???

13) P6L3, strength? is it density?

14) P6L19, relative error 30% is acceptable?

15) P6L24-25, give more literature to support

---

## Referee Comment (RC2) · Anonymous Referee #2 · 4 Feb 2018

Review: Effectiveness of low impact development for urban inundation risk mitigation under different scenarios: a case study in Shenzhen, China

J. Wu, R. Yang and J. Song

In this study a pipe drainage model (SWMM) coupled to a surface flow model is used to run test scenarios with different measures (green roofs and permeable pavements)

[Figure]

to mitigate inundation in a city after heavy rainfall. For this purpose, a test area of about 37 km$^2$ in a city in China is used. Different percentages of green roof area and permeable pavement are considered to compare maximal inundation depth, inundation area and inundation time between the different scenarios. It is concluded that both measures help to mitigate inundation. The gain is high if 25 percent of roofs are green and 25 percent of the pavement are permeable (compared to zero), while the change is not that much more when the percentage is increased further. It is also concluded that combines measures are more helpful than one measure alone with a higher percentage.

The topic is very interesting and relevant and it is in the focus of NHESS. The abstract of the manuscript is very motivating. However, the manuscript falls a bit short of what is announced. One expects that one learns about the coupling strategy of the two models. However, this part is not convincing (outlined below) and should be tuned down. The case studies with this model are sound. However, the study is somewhat superficial. There is hardly any discussion about the findings. For this reason, it is not clear how far these results are representative for other urban areas and if they could be transferred to other sites. Also, there is no discussion about limits of the study. Not much is learned about the processes that cause the effects that are described. This makes it also difficult to estimate if the results are transferrable or specific for the test case.

- Page 3, lines 16-18, 'However, we find that few researchs use....': I do not think that this is right. Dynamic models have maybe not been used much to test the measures for mitigation by permeable pavement and green roofs, but such coupled models (1d pipe drainage network model and 2d surface flow model) exist and are used for urban flood management (just a few arbitrarily chosen examples: R. Loewe, C. Ulrich, N.Sto. Domingo, O. Mark, A. Deletic and K. Arnbjerg-Nielsen (2017): Assessment of urban pluvial flood risk and efficiency of adaptation options through simulations - A new generation of urban planning tools. Journal

of Hydrology 550, 355-367. B. Russo, D. Sunyer, M. Velasco and S. Djordjevic (2015): Analysis of extreme flooding events through a calibrated 1d/2d coupled model: the case of Barcelona (Spain). Journal of Hydroinformatics 17(3), 473-490. M. J. Burns, J. E. Schubert, T. D. Fletcher and B. F. Sanders (2015): Testing the impact of at-source stormwater management on urban flooding through a coupling of network and overland flow models. WIREs Water 2. 291-300).

- 2.2 Data, part 4: What were the criteria for removing nodes and pipelines? A reduction from 4502 to 597 pipelines and from 1175 to 653 nodes seems a bit more that deleting some redundant and incorrect data. How was it tested that the data were redundant?

- 2.4 Coupling the SWMM/IFMS Urban models: As written above, I think that one does not learn much about the coupling. Also, Figure 4 does not help in this respect. One just learns that the models were coupled. But how were they coupled? Is inflow and outflow from and to manholes possible? What were the criteria for inflow and outflow? What was the spatial resolution of the geometry of a street? What timesteps were chosen for coupling? Either more discussion about the coupling is needed, which means that one also need to know more about the numerical schemes used for the two different models, or it does not make sense to have a section for this part.

- Page 5, lines 18-20: This sentence is unclear. Also: What is innovative about the coupling?

- Page 5, line 26: Why was a geostatistical method (Kriging) used for interpolation? I do not see the connection to geostatistics for a digital elevation model in a city.

- Page6, top: Please explain why green roofs should not be possible in a dense construction land.

- Modeling part (Section 2): Please explain how green roofs and permeable pavements are realized in the model. I assume that a storage for a roof area is assigned (or an existing one is increased) and that there is a soil compartment which gets a connection to the paved area if the pavement is permeable. As this is the key process that is here investigated, I think it is necessary to outline these things (and it is not enough to refer to the manuals of the models).

- Page 7, top: Please explain why the classification in hazard levels is made. What can be learned from the classification? It is written that the changes of inundation level are different for the different classes. But what does one make out of this fact? More discussion about consequences would be useful.

- Page 7, line 4: Please name scenarios 1 to 4.

- Figure 6: What is meant by percentage GR and PP? Both with the same percentage?

- Page 7, lines 14-18: I do not see where this conclusion comes from. Is this concluded from the numbers in Table 4? What is here meant by performance? Reduction of maximum inundation? This paragraph needs clarification.

- Page 7, lines 28-31: Again it is not clear where these numbers come from. I do not find it in the Figures. In Figure 6, the single 100 percent cases are not shown.

- Page 8, line 11: This needs explanation. Why is it difficult to mitigate? Is the reason the topography? I think that such a statement needs to be more specific.

- Page 8, lines 12-13: How can one see in these figures that the infrastructure is not perfect? And what infrastructure is here meant and how does it influence the inundation? Also: How can one see from these figures that the LID practices are not perfect? In which sense are they not perfect?

- Page 8, lines 13-15: I could not follow the reasoning. Why does the mitigation of short-time inundation areas lead to an increase in the average inundation time with LID measures? Is here something meant along the lines: If a storage due to green roofs helps to keep water back, leading to less inundation depth, the storage will at the same time lead to a longer inundation time (it holds the water back, but releases it eventually)? I am just guessing and I think this needs a better explanation.

- Section 4.1, Comparison of permeable pavement and green roofs: What is the reasoning of the different effects? This should be explained based on the mitigation mechanisms. The last sentence sounds a bit strong. I do not think that one test case can use as a proof, if no general reasoning is given for the different performances.

- Page 8, line 29: I would be a bit more careful with the word 'comprehensively'. The paper shows one case study. I do not think that this is a comprehensive exploration of inundation mitigation in an urban watershed.

- Page 9, lines 10-14: As before, I do not see the point about infrastructure. How is poor infrastructure reflected in the model? If not at all: How can one draw any conclusions about this point from a modeling study that does not capture this effect? If yes: What exactly is meant by poor infrastructure and how is this realized in the model?

- Page 22-23: Maybe this sentence is only not formulated well. But I do not see how from this study one could see anything about landscape patterns ('we find that the...' sounds as if it is a conclusion from this study). The landscape patterns are not discussed, so one cannot conclude about this point. For this reason, I can also not see how 'this provides a new perspective'. Or is here simply meant that this point should be studied in the future? In this case the sentences need to be reformulated.

- Conclusions: I think it should be mentioned that the findings in this study apply to the one test case considered. It is not clear if the results are more general and could be transferred to other sites. In particular: Numbers can certainly not be transferred.

---

## Author Comment (AC1) · 27 Feb 2018

Dear Referee #1:

Thank you for the valuable comments. We have carefully read all the comments, and our responses to your questions are listed below. We greatly appreciate your time and efforts to help us to improve our manuscript for further revision and publication.

**General comments**

This study sought to evaluate the impacts of LID practices on urban inundation at a watershed scale in China. Extensive modeling was used to assessed various LID implementation scenarios with a hydrodynamic inundation model, which coupled SWMM and IFMS Urban models. The study is interesting and will contribute to the understanding of LID effectiveness related to flood reduction. However, the scientific quality and presentation quality were poor. First, English in this paper is poor. some contents are difficult to understand. I would strongly recommend the editing by an experienced or even better native English speaker. Next there some major and obvious weakness in methodology and results. I listed them below. Also, it requires lots of improvements in other sections.

Re: Thank you for your recognition for our research. The language of this paper has been proofread by "Editage", a worldwide professional editing company. Since it is still difficult to be understood, we will invite one or two native speakers in our research area to proofread it again. I am so sorry that the poor expression of this paper make you confused. Therefore, in the following part, we will try our best to explain your questions and we also have modified them in the revised manuscript. Thanks again for your patient reading and valuable advices.

**Introduction:**

1. Review should be correct. In page 3 line 16, "we find that few researches use hydrodynamic models, like SWMM ...". In fact, there are many studies of SWMM in LID field, especially in 2017 in China. Also, the introduction is very universal, does not clearly lead to the specific content of the manuscript and is missing a central theme. For readers to quickly catch your contribution, it would be better to highlight major difficulties and challenges, and your original achievements to overcome them in a clearer way.

Re: Our statement about coupled models is imprecise. Thank you for your kind reminder, and we will modify the expression and here is the revised version of the last paragraph of Introduction.

It is noteworthy that peak flow reduction, runoff reduction, and hydrograph delay are widely used indexes when evaluating the performance of LID practices (Ahiablame and Shakya, 2016; Qin et al., 2013; Zhang et al., 2016). However, these indexes are not very intuitive and how LID practices perform on urban inundation is more beneficial to local residents, such as providing guides for their travel behaviours.Indeed, some 1D-2D models have been applied for flood management such like ESTRY-TUFLOW (Fewtrell et al., 2011), InfoWorks ICM (Russo et al., 2015) and MIKE FLOOD (Loewe et al., 2017). However, most of these models are not free that limits their applications, and the open-source model (like SWMM) with LID module

that can be coupied to simulate the urban inundation is needed in recently researches (Burns et al., 2015, Wu et al., 2017, Hu et al., 2017).

Therefore, the goal of this study is to demonstrate through a case study the effectiveness of LID practices to mitigate urban inundation in an urban watershed. The specific objectives were to (1) establish a 1D-2D hydrodynamic model coupled SWMM and IFMS Urban; and (2) evaluate the effectiveness of LID practices under different scenarios and hazard levels; and (3) explore the effiency of designed scenarios that related to the effectiveness of LID practices and the proportion of implementation areas. This study hopes to enrich the inundation mitigation research of LID on an urban watershed scale and provide some references to urban stormwater management and inundation mitigation for local government.

**Materials and methodology:**

1. Why you selected these two events? Were they have special characteristics?

Re: We chose two rainstorm events (11 May 2014 and 10 May 2016) for model simulation. On the one hand, the rainfall data and patterns for these two events are available that can be used for model calibration and validation. One the other hand, the increase in the frequency and intensity of urban flooding events associated with these types of rainstorm events (http://www.chinanews.com/gn/2014/06-10/6260988.shtml) highlights the need for these types of rainstorm events. So we think the two events have representations to carry out the research.

2. How you downscaled the dem resolution? The bias from downscaling was corrected?

Re: We resampled the DEM using Resample tool in ArcGIS 10.1. The aim is to compare the accuracy with the results of Kriging interpolation and we did not use the downscaled DEM for model simulation. This sentence seems useless and we will delete in the revised manuscript.

3. land use area should be described as well as the implementation area of each LID scenario

Re: Revised as requested.

4. Is there discharge Data for SWMM calibration?

Re: According to our detailed investigation on local government agencies, there is no discharge data that can be used for our model calibration. Indeed, lacking hydrologic data is a common problem for this type of research and it is even worse in China. Notheless, using inundation data to calibrate the model is an alternative and wide accepted way to calibrate models, and it has been applied in Hu et al. (2017) and Wu et al. (2017).

5. Why you coupled SWMM and IFMS Urban models? What the advantages compared with others? This study discussed inundation depth, area and time. There three indices could be got from some 2D inundation model. As I know, the outputs of SWMM are outflow, peak flow, flood volume, etc. This study didn't mention any of them. So, why you need SWMM?

Re: The reasons for choosing and coupling these two models are not clearly stated in section 2.3 and 2.4 of our original paper, we have made some descriptions to revise it in our revised manuscript:

SWMM is a 1D rainfall-runoff model which can use the given hydrology data and hydrodynamics to simulate the quantity and quality of rainfall-runoff. Nontheless, when the node overflow occurs, SWMM cannot simulate the spatial and temporal distributions of surface inundation, but the IFMS Urban can using 2D shallow water equations. However, the simulations of IFMS Urban must base on the simulated results of SWMM. So we coupled these two models to realize the simulation on the spatial and temporal distributions of surface inundation. And the outputs of the coupled model are inundation depth, inundation areas and inundation time. Indeed, we are more concerned about the results of surface inundation, and the outputs of SWMM are not showed in this research.

SWMM is an open-source model and it has been widely used to simulate the hydrologic performance of LID practices. IFMS Urban has great compatibility with ArcGIS and SWMM, and it can simulate surface inundation using DEM. What's more, the process of data conversion and model coupling are accomplished in IFMS Urban, and it doesn't need any other software programming, which is convenient for researchers and non-expert users.

**Results:**

1. The results for hazard level seem very sensitive to the thresholds chosen. Please give information on the thresholds chosen.

Re: The main basis for the thresholds is according to the relationship between vehicle speed and inundation depth researched in Su et al. (2016). Comparing their results with the study status, we set the three hazard levels for this research. Indeed, different thresholds might inform the results for hazard level and researches on more accurate thresholds are needed in future studies. We will put it in the Limitations and future studies in  the revised manuscript.

2.  Results are contradicted. The authors reported on Page 7 line 11-12 "the reduction effects become more evident as hazard level increases", "the roles of LID practices with respect to urban inundation mitigation are less obvious at High levels than thoseat Low levels". So which one is correct?

Re: From line 4-5 on page 7 of our manuscript, our research results show that for the High levels, the *depth reduction* after the construction of LID practices is from 0.11 m to 0.19 m (greater than that for the Low levels) and the *depth reduction rates* are from 22 % to 40 % (lower than those for the Low levels) under Scenarios 1 to 4. We didn't express clearly about the results in our original manuscript but we will improve it in the revised manuscript.

3. please show the spatial distribution of reductions in inundation depths instead of average reduction

Re: Figure 5 shows the spatil distribution of reductions in maximum inundation depths of the study area. And from this figure we can see the spatial changes of inundation depth in different scenarios. So we didn't show the spatial distribution of reductions in inundation depths.

4. please give more information of PP and GR implementation area, otherwise, you cannot say PP performs better than GR

Re: Thanks to point out our careless on the information missing. Data information is as follows:

The available implementation area of PP and GR is 5.95 km$^2$ and 8.92 km$^2$, respectively. The depth reduction rates of 100% PP are 67%, 38% and 23% at Low, Medium and High levels, and the depth reduction rates of 100% GR are 61%, 31% and 21% in three hazard levels. The area reduction rates of 100% PP are 37%, 65% and 67% at Low, Medium and High levels, and the area reduction rates of 100% GR are 32%, 56% and 67% at three hazard levels. Although the implementation area of PP is smaller than GR, the effectiveness of PP on urban immunation mitigation is greater than GR. So we say that PP performs better than GR in this study.

5. in section 3.3 you said the reduction in inundation area under High level was more obvious, but in section 3.1, reduction in inundation depth was less obvious. Please explain.

Re: Poor expression makes this part confusing to be understood but we have improved the expression in the revised manuscript. From the simulated results shown in section 3.2, the ***depth reduction*** after the construction of LID practices is greater but the ***depth reduction rate*** is lower under the High levels compared to Low levels (***question 2, Results***).

In section 3.3, the ***area reduction rate*** is greater under High level compared to other hazard levels (line 22 on page 7). This is because that after the construction of LID practices, in High level, the inundation depth has been decreased and most inundation areas are downgraded from High level to Medium or Low levels, but most inundation areas heavn't been eliminated which make the ***depth reduction rate*** lower than other levels. This is the reason why the ***depth reduction rate*** is lower and the ***area reduction rate*** is greater in High level compared with other levels.

6. please show the spatial distribution of reductions in inundation time instead of average

Re: Through the analysis of inundation depth and inundation area, we can draw the conclusions of this study approximately, and the analysis of inundation time comfirm effectiveness of LID practices from another aspect. Considering from the full text, inundation time is not the key point in this study, so we didn't show the map of inundation time. If necessary we will discuss it further in the revised manuscript.

7. one of the key points in your study is to compare the differences of all scenarios at three hazard levels not to find the differences among three hazard levels

Re: Indeed we both consider the two groups of comparisons in results. From Figure 6 we can see that as the propotion rises from 25% to 100%, the ***depth reduction rate*** (a) and ***area reduction rate*** (b) both increase in

the Low, Medium and High levels. It is clear that the reduction rate grows slowly associated with the increasing of proportion of LID implementation from 25% to 100%, which means the efficiency of LID implementation decreases from Scenario 1 to Scenario 4. To better describe the phenomenon, we will built a cost-effectiveness indicator (RPI) in the revised manuscript:

$$\text{RPI} = \frac{R}{P}$$

R means reduction rate of inundation depth and inundation areas, and P means the proportion of LID implementation. From Table 6 we can see that the RPIs of 25% PP+25% GR are always higher than the other scenarios while higher RPI indicates higher efficiency. From the comparisons, we can conclude that the simple increase of the proportion of LID implementation cannot necessarily contribute to the higher efficiency. Finally, we find that the efficiency of 25 % PP + 25 % GR is higher than other scenarios in this study. This indicates that we should not only consider the effectiveness but also the cost of LID practices in the construction of "Sponge City".

Table 6 RPI under differrent scenarios.

| | | 25%PP+25%GR | 50%PP+50%GR | 75%PP+75%GR | 100%PP+100%GR |
|---|---|---|---|---|---|
| Maximum inundation depth | | 0.64 | 0.44 | 0.35 | 0.29 |
| Average inundation depth | Low | 2.40 | 1.48 | 1.05 | 0.80 |
| | Medium | 1.08 | 0.86 | 0.68 | 0.54 |
| | High | 0.88 | 0.60 | 0.48 | 0.40 |
| Average inundation areas | Low | 1.23 | 0.87 | 0.68 | 0.53 |
| | Medium | 2.22 | 1.37 | 0.97 | 0.75 |
| | High | 2.86 | 1.62 | 1.14 | 0.90 |

**Discussion:**

1) The discussion is lacking depths. What are the same and different points comparing your study and others? What you studied from this research.

Re: Compared to the existing studies about LID, this study tries to explain the cost-effectiveness of LID for urban inundation risk mitigation. Moreover, this study focuses on the cost-effectiveness changes in different hazard levels under different scenarios.

1. The effectiveness of PP for urban inundation mitigation performs better than that of GR in this research. This conclusion might be different in other regions because of the differences of LID parameters, implementation area, spatial pattern, rainfall intensity, rainfall frequency and other factors. But it gives a reference for local residents and policy-maker that PP might be a good choice for local areas because of the great effectiveness and the large potential for reconstruction in the built-up region (PP could be gradually applied in roads and parking lots, while GR is hard to implement in density construction lands, especially in the urban villages);

2. Through the analysis in section 3.2 and 3.3, we can find that in High level, the inundation depth has been decreased and most inundation areas are downgraded from High level to Medium or Low levels, but most

high inundation hazard areas heavn't been eliminated and the ***depth reduction rate*** is lower than other levels. This indicates that LID practices can only ease the inundation depth and downgrade the inundation hazard level in High level. And some other methods of stormwater management should be used together to deal with severe waterlogging in High level areas;

3. Through the analysis in ***question 7, Results***, we find that the RPI decreases as the proportion of LID implementation increases from Scenario 1 to Scenario 4 and the efficiency of 25 % PP + 25 % GR is higher than other scenarios in this study. This indicates that the simple increase of the proportion of LID implementation cannot necessarily contribute to the higher efficiency, and we should not only consider the effectiveness but also the cost of LID practices in the construction of "Sponge City". These findings may provide some suggestions for LID designs in other regions.

2) The discussion on cost-effectiveness completely fell from the sky on page 9 line 3. You neither present how the costs were estimated nor discussed them in the Results.

Re: The main difference among scenarios from Scenario 1 to Scenario 4 is the proportion of LID implementation, and the cost will be higher as the proportion of LID implementation increases. Therefore, we develop a cost-effectiveness indicator (RPI) to discuss on the efficiency of LID practices (***question 7, Results***). We will add these descriptions in the revised manuscript.

3) In page 9 line 22, "we also find that spatial distribution of landscape patterns ...". This information completely fell from the sky. You neither present them in the Results.

Re: Thanks for pointing out the expression problem that these results are from Kim and Park (2016) and Giacomoni and Joseph (2017), and we will modify it in the revised manuscript.

4) You reported 25% of PP and GR had the highest efficiency. Is it correct? Do you consider the effect of rainfall intensity and frequency? LID effectiveness is highly related to rainfall intensity and frequency.

Re: You made a very constructive suggestion. We did find that rainfall intensity and frequency will influence effectiveness of LID. However, this study focuses on the trade-offs between implementation cost and effectiveness of LID practices, and we did not change the rainfall intensity or other factors in this study. In our research, once-in-100-years heavy rain happened on 11 May 2014 (144.9 mm) is selected to simulate the urban inundation situation. Because we find heavy rain of this intensity attacks Shenzhen almost very year associated with climate change. In this research, place-based references are provided for the policy-makers, and we do not suppose all the findings of this research can be directly transferrable to other places, cities even countries but the analytical methods and the efficiency analysis.

**Specific comments**

1) Page (P) 1 line (L) 15-19, too long to understand.

Re: This study proposes a hydrodynamic inundation model, coupling SWMM (Storm Water Management Model, 1D) and IFMS Urban (Integrated Urban Flood Modeling System, 2D), to assess the effectiveness of LID practices under different scenarios and hazard levels. The results are shown as follows.

2) P1L25, considering cost-effectiveness, you don't give any information on it.

Re: The information about cost-effectiveness is mentioned above (*question 2, Discussion*).

3) P2L4: what are secondary disasters? it is better to delete.

Re: Amended as requested.

4) P3L18-20, there some studies on this topic, please review them.

Re: Amended as requested.

5) P3L29, give rainfall information from April to September.

Re: April to September marks the rainy season in Shenzhen. There are 38 rainstorm days (95% of the whole year) in 2017 and the average rainfall is 170-350 mm every month during this period.

6) P4L5-10. simplify the description.

Re: The study site is located in Guangming New District of Shenzhen, China (Fig. 1). The total area of this study site is 37.68 km$^2$ with 69.8 % of it is the construction land. Because of the intensive inundation disasters, Guangming New District was selected as the first pilot area for LID practices in Shenzhen in October 2011. Therefore, there is a need to research the effectiveness of LID practices on urban inundation mitigation in this area.

7) P4L12, delete"needed for modeling".

Re: Amended as requested.

8) P4L18-19, how to do.

Re: We resampled the DEM using Resample tool in ArcGIS 10.1.

9) P4L20-22, improve.

Re: The reason why we choose the two events is mentioned above (*question 1, Materials and methodology*) and we will improve it in the revised manuscript.

10) P5L1-3, improve.

Re: We have reorganized section 2.3 and 2.4 in the revised manuscript.

11) give clear information on the model.

Re: The detailed information about the model is introduced above (*question 5, Materials and methodology*) and we will improve it in the revised manuscript.

12) P5L31-32, "we found ..."???

Re: We have not explained the details for this part. In fact, there are some special attributes for buildings on the dense construction land in our research area. Through the detailed urban planning and field investigations of our research area, we found the 80% of the residential lands are urban villages, desnsely constructed on construction lands. The structures and shapes of roofs for urban villages are diversity which makes it difficult to build green roofs on them. More important, the complex owenership and financing pathways which also

make it difficult to construct the green roofs for the dense construction lands in our research area. Thereforce, we temporarily didn't set green roof in the dense construction land in this study.

13) P6L3, strength? is it density?

Re: We will instead "Construction strength" of "construction density" here.

14) P6L19, relative error 30% is acceptable?

Re: Lacking observation data is a universal problem in model simulation, and some models did not have a calibration (Hu et al., 2017). In this study, the relative error of calibration seems a little high, while the relative errors of validation are 5-20%, which is met the requirements of the Standard for Hydrologic Information and Hydrologic Forecasting in China (GBT_22482-2008). If there are more detailed inundation records, the model can be further improved in the future study. We will discuss the limitation in section 4.4 Limitations and future studies.

15) P6L24-25, give more literature to support

Re: Amended as requested.

**References**

Ahiablame, L., and Shakya, R.: Modeling flood reduction effects of low impact development at a watershed scale, J Environ Manage, 171, 81-91, 10.1016/j.jenvman.2016.01.036, 2016.

Burns, M. J., Schubert, J. E., Fletcher, T. D., and Sanders, B. F.: Testing the impact of at-source stormwater management on urban flooding through a coupling of network and overland flow models, Wiley Interdiscip. Rev.-Water, 2, 291-300, 10.1002/wat2.1078, 2015.

Fewtrell, T. J., Neal, J. C., Bates, P. D., and Harrison, P. J.: Geometric and structural river channel complexity and the prediction of urban inundation, Hydrological Processes, 25, 3173-3186, 10.1002/hyp.8035, 2011.

Giacomoni, M. H., and Joseph, J.: Multi-Objective Evolutionary Optimization and Monte Carlo Simulation for Placement of Low Impact Development in the Catchment Scale, J. Water Resour. Plan. Manage.-ASCE, 143, 15, 10.1061/(asce)wr.1943-5452.0000812, 2017.

Hu, M., Sayama, T., Zhang, X., Tanaka, K., Takara, K., and Yang, H.: Evaluation of low impact development approach for mitigating flood inundation at a watershed scale in China, J Environ Manage, 193, 430-438, 10.1016/j.jenvman.2017.02.020, 2017.

Kim, H. W., and Park, Y.: Urban green infrastructure and local flooding: The impact of landscape patterns on peak runoff in four Texas MSAs, Applied Geography, 77, 72-81, 10.1016/j.apgeog.2016.10.008, 2016.

Lowe, R., Urich, C., Domingo, N. S., Mark, O., Deletic, A., and Arnbjerg-Nielsen, K.: Assessment of urban pluvial flood risk and efficiency of adaptation options through simulations - A new generation of urban planning tools, Journal of Hydrology, 550, 355-367, 10.1016/j.jhydrol.2017.05.009, 2017.

Qin, H. P., Li, Z. X., and Fu, G.: The effects of low impact development on urban flooding under different rainfall characteristics, J Environ Manage, 129, 577-585, 10.1016/j.jenvman.2013.08.026, 2013.

Russo, B., Sunyer, D., Velasco, M., and Djordjevic, S.: Analysis of extreme flooding events through a calibrated 1D/2D coupled model: the case of Barcelona (Spain), J. Hydroinform., 17, 473-491, 10.2166/hydro.2014.063, 2015.

Su, B., Huang, H., and Li, Y.: Integrated simulation method for waterlogging and traffic congestion under urban rainstorms, Natural Hazards, 81, 23-40, 10.1007/s11069-015-2064-4, 2016.

Wu, X., Wang, Z., Guo, S., Liao, W., Zeng, Z., and Chen, X.: Scenario-based projections of future urban inundation within a coupled hydrodynamic model framework: A case study in Dongguan City, China, Journal of Hydrology, 547, 428-442, 10.1016/j.jhydrol.2017.02.020, 2017.

Zhang, X., Guo, X., and Hu, M.: Hydrological effect of typical low impact development approaches in a residential district, Natural Hazards, 80, 389-400, 10.1007/s11069-015-1974-5, 2016.

---

## Author Comment (AC2) · 27 Feb 2018

Dear Referee #2:

Thank you for the positive comments and constructive suggestions on this paper, which we fully taken into account in the revised version of the paper. In the supplement we address and reply to the questions below.

1. Page 3, lines 16-18, 'However, we find that few researchs use...': I do not think that this is right. Dynamic models have maybe not been used much to test the measures for mitigation by permeable pavement and green roofs, but such coupled models (1d pipe drainage network model and 2d surface flow model) exist and are used for urban flood management (just a few arbitrarily chosen examples: R. Loewe, C. Ulrich, N.Sto. Domingo, O. Mark, A. Deletic and K. Arnbjerg-Nielsen (2017): Assessment of urban pluvial flood risk and efficiency of adaptation options through simulations - A new generation of urban planning tools. Journal of Hydrology 550, 355-367. B. Russo, D. Sunyer, M. Velasco and S. Djordjevic (2015): Analysis of extreme flooding events through a calibrated 1d/2d coupled model: the case of Barcelona (Spain). Journal of Hydroinformatics 17(3), 473- 490. M. J. Burns, J. E. Schubert, T. D. Fletcher and B. F. Sanders (2015): Testing the impact of at-source stormwater management on urban flooding through a coupling of network and overland flow models. WIREs Water 2. 291-300)

Re: Our statement about coupled models is imprecise. Thank you for your kind reminder, and we will modify the expression and here is the revised version of the last paragraph of Introduction.

It is noteworthy that peak flow reduction, runoff reduction, and hydrograph delay are widely used indexes when evaluating the performance of LID practices (Ahiablame and Shakya, 2016; Qin et al., 2013; Zhang et al., 2016). However, these indexes are not very intuitive and how LID practices perform on urban inundation is more beneficial to local residents, such as providing guides for their travel behaviours.Indeed, some 1D-2D models have been applied for flood management such like ESTRY-TUFLOW (Fewtrell et al., 2011), InfoWorks ICM (Russo et al., 2015) and MIKE FLOOD (Loewe et al., 2017). However, most of these models are not free that limits their applications, therefore the open-source model (like SWMM) with LID module that can be coupied to simulate the urban inundation is needed in recently researches (Burns et al., 2015, Wu et al., 2017, Hu et al., 2017).

Therefore, the goal of this study is to demonstrate through a case study the effectiveness of LID practices to mitigate urban inundation in an urban watershed. The specific objectives were to (1) establish a 1D-2D hydrodynamic model coupled SWMM and IFMS Urban; and (2) evaluate the effectiveness of LID practices under different scenarios and hazard levels; and (3) explore the effiency of designed scenarios that related to the effectiveness of LID practices and the proportion of implementation areas. This study hopes to enrich the inundation mitigation research of LID on an urban watershed scale and provide some references to urban stormwater management and inundation mitigation for local government.

2. 2.2 Data, part 4: What were the criteria for removing nodes and pipelines? A reduction from 4502 to 597 pipelines and from 1175 to 653 nodes seems a bit more that deleting some redundant and incorrect data. How was it tested that the data were redundant?

Re: Indeed, the actural drainage networks are compulsory and substantial. Nontheless, SWMM cannot accurately simulate when the data is huge. Besides, after the data conversion process for applying into the SWMM, some overlaps and break points for the pipelines are generated, which makes lots of nodes and pipelines useless. Therefore, we have to simply the drainage data for model building and the criteria shown below:

a. Add nodes when the pipeline is too long;

b. Keep or add the corner nodeschanging diameter nodes, or large variation range of slope nodes;

c. Keep the parallel pipelines and nodes on both sides of the roads;

d. Delete the useless nodes and pipelines in this model.

3. 2.4 Coupling the SWMM/IFMS Urban models: As written above, I think that one does not learn much about the coupling. Also, Figure 4 does not help in this respect. One just learns that the models were coupled. But how were they coupled? Is inflow and outflow from and to manholes possible? What were the criteria for inflow and outflow? What was the spatial resolution of the geometry of a street? What timesteps were chosen for coupling? Either more discussion about the coupling is needed, which means that one also need to know more about the numerical schemes used for the two different models, or it does not make sense to have a section for this part.

Re: SWMM is a 1D rainfall-runoff model which use the given hydrology data and hydrodynamics to simulate the quantity and quality of rainfall-runoff. Notheless, when the node overflow occurs, SWMM can't simulate the spatial and temporal distributions of surface inundation, but the IFMS Urban can using 2D shallow water equations. However, the simulations of IFMS Urban must base on the simulated results of SWMM. So we coupled these two models to realize the simulate on the spatial and temporal distributions of surface inundation. What's more, the process of data conversion and model coupling are all accomplished in IFMS Urban, and it doesn't need other software programming or specialized knowledge, which is convenient for researchers and non-expert users. So we don't want to make it complicated or list algorithm and formula in this part. The spatial resolution of the geometry of a street is 15 m. The timestep of calculation is 10 s and the timestep of output is 200 s.

4. Page 5, lines 18-20: This sentence is unclear. Also: What is innovative about the coupling?

Re: SWMM is a 1D hydrodynamics model which can simulate the quantity and quality of rainfall-runoff but it can't simulate the urban inundation, while the IFMS Urban is a 2D model which can simunate the urban inundation but it must base on the results of SWMM. Through coupling, we build a 1D-2D hydrodynamic model that can simulate the spatial and temporal distributions of surface inundation. Based on this coupled model, we can evaluate the effectiveness of LID from inundation depth, inundation area

and inundation time. And this coupled model both takes in the advantages of SWMM and IFMS Urban (open-source, free, great compatibility with ArcGIS and 2D inundation simulation), which is convenient for researchers and non-expert users.

5. Page 5, line 26: Why was a geostatistical method (Kriging) used for interpolation? I do not see the connection to geostatistics for a digital elevation model in a city.

Re: We need DEM when building the 2D model. However, the accuracy of DEM production from Geospatial Data Cloud can not meet our demand (for example, 6 m, 13 m). However, the high accuracy DEM is confidential and difficult to obtain in China. Alternatively, we find the ground elevation of nodes in pipe network data has a higher accuracy (for example, 6.588 m, 13.483 m), and the nodes on the roads are relatively dense. So we use a geostatistical method (Kriging) to get a high accuracy DEM of the roads with the elevation data of nodes on the roads.

6. Page6, top: Please explain why green roofs should not be possible in a dense construction land.

Re: We have not explained the details for this part and thank you for your kind reminder. In fact, there are some special attributes for buildings on the dense construction land in our research area. Through the detailed urban planning and field investigations of our research area, we found the 80% of the residential lands are urban villages, desnsely constructed on construction lands. The structures and shapes of roofs for urban villages are diversity which makes it difficult to build green roofs on them. Thereforce, we temporarily didn't set green roof in the dense construction land in this study.

7. Modeling part (Section 2): Please explain how green roofs and permeable pavements are realized in the model. I assume that a storage for a roof area is assigned (or an existing one is increased) and that there is a soil compartment which gets a connection to the paved area if the pavement is permeable. As this is the key process that is here investigated, I think it is necessary to outline these things (and it is not enough to refer to the manuals of the models).

Re: The simulation designs and parameter setting for PP and GR are listed in Table 1 of our paper, which are strictly desinged according to the manual of SWMM and some highly cited studies of LIDs (Ahiablame and Shakya, 2016; Chui et al., 2016; Kong et al., 2017; Qin et al., 2013).

8. Page 7, top: Please explain why the classification in hazard levels is made. What can be learned from the classification? It is written that the changes of inundation level are different for the different classes. But what does one make out of this fact? More discussion about consequences would be useful.

Re: Through the classification in hazard levels, we can explore the effectiveness of LID practices in different hazard levels, especial in the High level. Through the analysis in section 3.2 and 3.3, we can find that in the High levels, the inundation depth has been decreased (*depth reduction rates* are from 22% to 40

%) and most inundation areas are downgraded from High levels to Medium or Low levels (*area reduction rates* are from 71% to 90 %), but most inundation areas heavn't been eliminated and the *depth reduction rate* is lower than other levels (lower 38-40% than Low level). This indicates that LID practices can only ease the inundation depth and downgrade the inundation hazard level and can't thoroughly resolve the inundation problem in High level. And some other methods of stormwater management should be used together to deal with severe waterlogging at High level areas.

9. Page 7, line 4: Please name scenarios 1 to 4

Re: Amended as requested.

10. Figure 6: What is meant by percentage GR and PP? Both with the same percentage?

Re: The proportion means the percentage of the total available implementation areas of LID. Here the *percentage GR and PP* means the proportion of Scenario 1 to Scenario 4 (from 25% to 100%) in Figure 6.

11. Page 7, lines 14-18: I do not see where this conclusion comes from. Is this concluded from the numbers in Table 4? What is here meant by performance? Reduction of maximum inundation? This paragraph needs clarification.

Re: We did not put the data in the part that the *depth reduction rates* of 100% PP are 67%, 38% and 23% at Low, Medium and High levels, and the *depth reduction rates* of 100% GR are 61%, 31% and 21% at three hazard levels. Here the performance means the average depth reduction rate. We will reorganize this paragraph.

12. Page 7, lines 28-31: Again it is not clear where these numbers come from. I do not find it in the Figures. In Figure 6, the single 100 percent cases are not shown.

Re: We did not put the data in the part that the *area reduction rates* of 100% PP are 37%, 65% and 67% at Low, Medium and High levels, and the *area reduction rates* of 100% GR are 32%, 56% and 67% at three hazard levels. We will add the data in Figure 6.

13. Page 8, line 11: This needs explanation. Why is it difficult to mitigate? Is the reason the topography? I think that such a statement needs to be more specific.

Re: The topographical attributes, such as concaves and potholes, are easy to lead to some places got inundation on the road surfaces. If these places are not or not enough drainage pipes to drainage the rainwater, it is difficult for them to mitigate the influences of urban inundation even there are LIDs. Because of these long-time inundation time areas, the average inundation time increases 0.1 h after the implementation of LID practices (*question 15*).

14. Page 8, lines 12-13: How can one see in these figures that the infrastructure is not perfect? And what infrastructure is here meant and how does it influence the inundation? Also: How can one see from these figures that the LID practices are not perfect? In which sense are they not perfect?

Re: Here we want to explain why some places are difficult to mitigate (*question 13*). These sentences are not rigorous and we will modify them in the revised manuscript.

15. Page 8, lines 13-15: I could not follow the reasoning. Why does the mitigation of short-time inundation areas lead to an increase in the average inundation time with LID measures? Is here something meant along the lines: If a storage due to green roofs helps to keep water back, leading to less inundation depth, the storage will at the same time lead to a longer inundation time (it holds the water back, but releases it eventually)? I am just guessing and I think this needs a better explanation.

Re: Indeed, this is because the statistical number of urban inundation areas are not the same before and after the implementation of LID practices. Here we want to explain why the average inundation time increases 0.1 h after the implementation of LID practices. Because of the implementation of LID practices, the inundation time has been decreased in all hazard levels. However, for the Low level some short-time inundation areas previously affected by surface runoff are freed from urban flooding after the construction of the LID projects, which makes the total number of inundation areas decreases after the implementation of LID. More important, the most freed areas are short-time inundation areas. Although LID practices make existing urban inundation areas' inundation time shorten, the statistical data suggest that the average of the lasting inundation areas' inundation duration is a little longer than that before LID practices. It is also suggests the great effectiveness of LID practices at Low level. We will modify the sentences in line 11-15 and make them clearer to understand.

16. Section 4.1, Comparison of permeable pavement and green roofs: What is the reasoning of the different effects? This should be explained based on the mitigation mechanisms. The last sentence sounds a bit strong. I do not think that one test case can use as a proof, if no general reasoning is given for the different performances.

Re: The available implementation area of PP and GR is 5.95 km$^2$ and 8.92 km$^2$, respectively. Although the implementation area of PP is smaller than GR, the effectiveness of PP on urban immunation mitigation is greater than GR in this study (*question 11, 12*). Except the differences of LID parameters, the reason of the different effects might be that PP is built both on low ad high construction lands, while GR is only built on low density construction lands. Indeed, the effectiveness of PP for urban inundation mitigation were different from studies (Qin et al., 2013, Ahiablame and Shakya, 2016, Zhang et al., 2016, Hu et al., 2017), and PP can not always perform better because that the effectiveness is depended on the parameters, implementation area, spatial pattern, rainfall intensity, rainfall frequency and other factors in different regions. Here we want to give a reference for local government that PP might be a good choice for local areas because of the great effectiveness and the large potential for reconstruction in the built-up region (PP

could be gradually applied in roads and parking lots, while GR is hard to implement in density construction lands, especially in the urban villages).

17. Page 8, line 29: I would be a bit more careful with the word 'comprehensively'. The paper shows one case study. I do not think that this is a comprehensive exploration of inundation mitigation in an urban watershed.

Re: We will delete the word.

18. Page 9, lines 10-14: As before, I do not see the point about infrastructure. How is poor infrastructure reflected in the model? If not at all: How can one draw any conclusions about this point from a modeling study that does not capture this effect? If yes: What exactly is meant by poor infrastructure and how is this realized in the model?

Re: The scentences in lines 10-14 are not rigorous. Indeed, we find that the efficiency decreases as the proportion of LID implementation increases from Scenario 1 to Scenario 4 and the efficiency of 25 % PP + 25 % GR is higher than other scenarios in this study. This indicates that the greater proportion of LID implementation might not lead to the higher efficiency, and we should not only consider the effectiveness but also the cost of LID practices in the construction of "Sponge City".

19. Page 22-23: Maybe this sentence is only not formulated well. But I do not see how from this study one could see anything about landscape patterns ('we find that the...' sounds as if it is a conclusion from this study). The landscape patterns are not discussed, so one cannot conclude about this point. For this reason, I can also not see how 'this provides a new perspective'. Or is here simply meant that this point should be studied in the future? In this case the sentences need to be reformulated.

Re: Thanks for pointing out the expression problem that these results are from Kim and Park (2016) and Giacomoni and Joseph (2017), and we will modify it in the revised manuscript.

20. Conclusions: I think it should be mentioned that the findings in this study apply to the one test case considered. It is not clear if the results are more general and could be transferred to other sites. In particular: Numbers can certainly not be transferred.

Re: This study is a simulation-based research on a local basis. Although the results cannot be transferable to other places directly, the analytical methods, including the coupling model, cost-effectiveness analysis during the sponge city construction can be transferable. We will list the main conclusions below:

1. The coupling model with SWMM and IFMS Urban can be applied to evaluate the effectiveness of LID for urban inundation risk mitigation and can be transferred to other sites.

2. The effectiveness of PP for urban inundation mitigation performs better than that of GR in this research. This conclusion might be different in other regions but it gives a reference for policy-maker on a local basis.

3. LID practices can only ease the inundation depth and downgrade the inundation hazard level but can't thoroughly resolve the inundation problem in High level. Therefore, some other methods of stormwater management should be used together to deal with severe waterlogging at High level areas.

4. The greater proportion of LID implementation might not lead to the higher efficiency, and we should not only consider the effectiveness but also the cost of LID practices in the construction of "Sponge City".

**References**

Ahiablame, L., and Shakya, R.: Modeling flood reduction effects of low impact development at a watershed scale, J Environ Manage, 171, 81-91, 10.1016/j.jenvman.2016.01.036, 2016.

Burns, M. J., Schubert, J. E., Fletcher, T. D., and Sanders, B. F.: Testing the impact of at-source stormwater management on urban flooding through a coupling of network and overland flow models, Wiley Interdiscip. Rev.-Water, 2, 291-300, 10.1002/wat2.1078, 2015.

Chui, T. F. M., Liu, X., and Zhan, W.: Assessing cost-effectiveness of specific LID practice designs in response to large storm events, Journal of Hydrology, 533, 353-364, 10.1016/j.jhydrol.2015.12.011, 2016.

Fewtrell, T. J., Neal, J. C., Bates, P. D., and Harrison, P. J.: Geometric and structural river channel complexity and the prediction of urban inundation, Hydrological Processes, 25, 3173-3186, 10.1002/hyp.8035, 2011.

Giacomoni, M. H., and Joseph, J.: Multi-Objective Evolutionary Optimization and Monte Carlo Simulation for Placement of Low Impact Development in the Catchment Scale, J. Water Resour. Plan. Manage.-ASCE, 143, 15, 10.1061/(asce)wr.1943-5452.0000812, 2017.

Hu, M., Sayama, T., Zhang, X., Tanaka, K., Takara, K., and Yang, H.: Evaluation of low impact development approach for mitigating flood inundation at a watershed scale in China, J Environ Manage, 193, 430-438, 10.1016/j.jenvman.2017.02.020, 2017.

Kim, H. W., and Park, Y.: Urban green infrastructure and local flooding: The impact of landscape patterns on peak runoff in four Texas MSAs, Applied Geography, 77, 72-81, 10.1016/j.apgeog.2016.10.008, 2016.

Kong, F. H., Ban, Y. L., Yin, H. W., James, P., and Dronova, I.: Modeling stormwater management at the city district level in response to changes in land use and low impact development, Environ. Modell. Softw., 95, 132-142, 10.1016/j.envsoft.2017.06.021, 2017.

Lowe, R., Urich, C., Domingo, N. S., Mark, O., Deletic, A., and Arnbjerg-Nielsen, K.: Assessment of urban pluvial flood risk and efficiency of adaptation options through simulations - A new generation of urban planning tools, Journal of Hydrology, 550, 355-367, 10.1016/j.jhydrol.2017.05.009, 2017.

Qin, H. P., Li, Z. X., and Fu, G.: The effects of low impact development on urban flooding under different rainfall characteristics, J Environ Manage, 129, 577-585, 10.1016/j.jenvman.2013.08.026, 2013.

Russo, B., Sunyer, D., Velasco, M., and Djordjevic, S.: Analysis of extreme flooding events through a calibrated 1D/2D coupled model: the case of Barcelona (Spain), J. Hydroinform., 17, 473-491, 10.2166/hydro.2014.063, 2015.

Wu, X., Wang, Z., Guo, S., Liao, W., Zeng, Z., and Chen, X.: Scenario-based projections of future urban inundation within a coupled hydrodynamic model framework: A case study in Dongguan City, China, Journal of Hydrology, 547, 428-442, 10.1016/j.jhydrol.2017.02.020, 2017.

Zhang, X., Guo, X., and Hu, M.: Hydrological effect of typical low impact development approaches in a residential district, Natural Hazards, 80, 389-400, 10.1007/s11069-015-1974-5, 2016.

---

## Author Response (AR1)

**RESPONSE TO THE EDITOR**

Dear Authors,

According to the comments of the two reviewers, of which one suggested "reject" and one "major revisions" your manuscript needs substantial further revisions before it can be reconsidered for publication. Particularly the method and result descriptions of your study need to be improved. Reviewers identify similar shortcomings of your manuscript, which need special attention: (i) the methodology e.g. coupling of the models is unclear, (ii) the description and discussion of your findings need to be improved, (iii) to which extent are the results specific for the case study and to which extent and how could they be transferred to other sites.

I ask you to revise your manuscript in accordance with all the comments and recommendations of each of the reviewers. When you have completed your revision, please submit your revised manuscript with the changes marked, and a detailed item-by-item response to each of the reviewer's comments.

Best regards

Heidi Kreibich

Dear Editor,

Thank you for your time and efforts for helping us improve our manuscript. We have major revised our manuscript and a summary of the revision is provided as the following.

The description of the two models and the process of model coupling were modified in section **2.3** and **2.4**. In section **2.3**, we introduced the advantages and disadvantages of the two models and explained the reason for model coupling. In section **2.4**, we re-wrote the process of model coupling hoping to make it more clear and understandable. We further improved the description and discussion by amending the most paragraphs and spliting the original section **4.2** into new section **4.2** and **4.3** to disscuss the effectiveness at different hazard levels and the cost-effectiveness of LID practices. In section **5**, we concluded that the practice of model coupling could be applied to other sites, and that most findings could be transferred to other sites except that PP were more effective for urban inundation mitigation than GR.

Details of the changes are presented in the revised manuscript. The detailed item-by-item response to the reviewers' comments are listed as the following. We deeply appreciate your consideration of our work. Please do not hesitate to contact us for any queries.

Best regards

On behalf of all the authors

Yang Rui

**RESPONSE TO THE REFEREE #1**

Dear Referee #1:

Thank you for the valuable comments. We have carefully read all the comments, and our responses to your questions are listed

5    below. We greatly appreciate your time and efforts to help us to improve our manuscript for further revision and publication.

**General comments**

This study sought to evaluate the impacts of LID practices on urban inundation at a watershed scale in China. Extensive modeling was used to assessed various LID implementation scenarios with a hydrodynamic inundation model, which coupled SWMM and

10   IFMS Urban models. The study is interesting and will contribute to the understanding of LID effectiveness related to flood reduction. However, the scientific quality and presentation quality were poor. First, English in this paper is poor. some contents are difficult to understand. I would strongly recommend the editing by an experienced or even better native English speaker. Next there some major and obvious weakness in methodology and results. I listed them below. Also, it requires lots of improvements in other sections.

15   Re: Thank you for your recognition for our research. The language of this paper has been proofread by "Editage", a worldwide professional editing company. Since it is still difficult to be understood, we will invite one or two native speakers in our research area to proofread it again. I am so sorry that the poor expression of this paper make you confused. Therefore, in the following part, we will try our best to explain your questions and we also have modified them in the revised manuscript. Thanks again for your patient reading and valuable advices.

20   *Change in manuscript: The language of this paper was proofread by two native speakers again and the expression of this paper was improved. Some confusing problems were modified in the revised manuscript.*

**Introduction:**

1. Review should be correct. In page 3 line 16, "we find that few researches use hydrodynamic models, like SWMM ...". In fact,

25   there are many studies of SWMM in LID field, especially in 2017 in China. Also, the introduction is very universal, does not clearly lead to the specific content of the manuscript and is missing a central theme. For readers to quickly catch your contribution, it would be better to highlight major diffculties and  challenges, and your original achievements to overcome them in a clearer way.

Re: Our statement about coupled models is imprecise. Thank you for your kind reminder, and we will modify the expression and

30   here is the revised version of the last paragraph of Introduction.

It is noteworthy that peak flow reduction, runoff reduction, and hydrograph delay are widely used indexes when evaluating the performance of LID practices  (Ahiablame and Shakya, 2016; Qin et al., 2013; Zhang et al., 2016). However, these indexes are not very intuitive and how LID practices perform on urban inundation is more beneficial to local residents, such as providing guides for their travel behaviours.Indeed, some 1D-2D models have been applied for flood management such like ESTRY-TUFLOW

35   (Fewtrell et al., 2011), InfoWorks ICM (Russo et al., 2015) and MIKE FLOOD (Loewe et al., 2017). However, most of these models are not free that limits their applications, and the open-source model (like SWMM) with LID module that can be coupied to simulate the urban inundation is needed in recently researches (Burns et al., 2015, Wu et al., 2017, Hu et al., 2017).

Therefore, the goal of this study is to demonstrate through a case study the effectiveness of LID practices to mitigate urban inundation in an urban watershed. The specific objectives were to (1) establish a 1D-2D hydrodynamic model coupled SWMM

40   and IFMS Urban; and (2) evaluate the effectiveness of LID practices under different scenarios and hazard levels; and (3) explore

the effiency of designed scenarios that related to the effectiveness of LID practices and the proportion of implementation areas. This study hopes to enrich the inundation mitigation research of LID on an urban watershed scale and provide some references to urban stormwater management and inundation mitigation for local government.

*Change in manuscript: We modified the paragraphs and listed the goals and specific objectives we want to achieve on Page 3, line 31 to Page 4, line 8.*

*"Peak flows reduction, runoff reduction, and hydrograph delays are widely used indexes for evaluating the performance of LID practices (Ahiablame and Shakya, 2016; Qin et al., 2013; Zhang et al., 2016). However, these indexes are not intuitive, and the performance of LID practices for urban inundation is more useful for local residents, such as providing a guide for their travel behaviour. Some 1D-2D models have been applied for flood management, such as ESTRY-TUFLOW (Fewtrell et al., 2011), InfoWorks ICM (Russo et al., 2015) and MIKE FLOOD (Loewe et al., 2017). However, most of these models have a cost, which limits their application, and an open-source model (like Storm Water Management Model, SWMM), with a LID module that can be coupled to simulate urban inundation, is needed (Burns et al., 2015, Hu et al., 2017, Wu et al., 2017).*

*The goal of this study was to evaluate the effectiveness of LID practices to mitigate urban inundation in an urban watershed using a case study. The specific objectives were to establish a 1D-2D hydrodynamic model that coupled SWMM and IFMS Urban, evaluate the effectiveness of LID practices under different scenarios and hazard levels, and explore the efficiency of the LID scenarios. We intended this study to enrich LID inundation mitigation research at the urban watershed scale and to provide a reference for urban stormwater management and inundation mitigation for local governments."*

**Materials and methodology:**

1. Why you selected these two events? Were they have special characteristics?

Re: We chose two rainstorm events (11 May 2014 and 10 May 2016) for model simulation. On the one hand, the rainfall data and patterns for these two events are available that can be used for model calibration and validation. One the other hand, the increase in the frequency and intensity of urban flooding events associated with these types of rainstorm events (http://www.chinanews.com/gn/2014/06-10/6260988.shtml) highlights the need for these types of rainstorm events. So we think the two events have representations to carry out the research.

*Change in manuscript: The two events are representative and have the complete records of rainfall and inundation data (Page 5, line 11 to 13).*

*"According to the integrity and availability of data, we chose two representative heavy rainstorm event datasets, 11 May 2014 and 10 May 2016 (Figure 3) for model simulation, which included the complete volume of rainfall every hour."*

2. How you downscaled the dem resolution? The bias from downscaling was corrected?

Re: We resampled the DEM using Resample tool in ArcGIS 10.1. The aim is to compare the accuracy with the results of Kriging interpolation and we did not use the downscaled DEM for model simulation. This sentence seems useless and we will delete in the revised manuscript.

*Change in manuscript: We deleted the useless paragraph on Page 5, line 9 –10.*

3. land use area should be described as well as the implementation area of each LID scenario

Re: Revised as requested.

*Change in manuscript: The total available area for PP and GR was on Page 7, line 17–18.*

4. Is there discharge Data for SWMM calibration?

Re: According to our detailed investigation on local government agencies, there is no discharge data that can be used for our model calibration. Indeed, lacking hydrologic data is a common problem for this type of research and it is even worse in China. Notheless,

using inundation data to calibrate the model is an alternative and wide accepted way to calibrate models, and it has been applied in Hu et al. (2017) and Wu et al. (2017).

*Change in manuscript: Based on inundation data, this model was calibrated on Page 8, line 2–16.*

5  5. Why you coupled SWMM and IFMS Urban models? What the advantages compared with others? This study discussed inundation depth, area and time. There three indices could be got from some 2D inundation model. As I know, the outputs of SWMM are outflow, peak flow, flood volume, etc. This study didn't mention any of them. So, why you need SWMM?

Re: The reasons for choosing and coupling these two models are not clearly stated in section 2.3 and 2.4 of our original paper, we have made some descriptions to revise it in our revised manuscript:

10  SWMM is a 1D rainfall-runoff model which can use the given hydrology data and hydrodynamics to simulate the quantity and quality of rainfall-runoff. Nontheless, when the node overflow occurs, SWMM cannot simulate the spatial and temporal distributions of surface inundation, but the IFMS Urban can using 2D shallow water equations. However, the simulations of IFMS Urban must base on the simulated results of SWMM. So we coupled these two models to realize the simulation on the spatial and temporal distributions of surface inundation. And the outputs of the coupled model are inundation depth, inundation areas and

15  inundation time. Indeed, we are more concerned about the results of surface inundation, and the outputs of SWMM are not showed in this research.

SWMM is an open-source model and it has been widely used to simulate the hydrologic performance of LID practices. IFMS Urban has great compatibility with ArcGIS and SWMM, and it can simulate surface inundation using DEM. What's more, the process of data conversion and model coupling are accomplished in IFMS Urban, and it doesn't need any other software

20  programming, which is convenient for researchers and non-expert users.

*Change in manuscript: We reorganized the paragraphs to introduce the advantages of the two models. The introduction of SWMM and IFMS Urban has been shown at section **2.3** ( Page 5, line 25–Page 6, line 5).*

*"SWMM is an open-source model that can simulate  dynamic runoff quantity and quality from urban areas, and it has been widely used to simulate the hydrologic performance of LID practices (Rossman, 2010; Wu et al., 2013). However, SWMM cannot simulate*

25  *the spatial and temporal distributions of surface inundation. Recently, some scholars have conducted simulationsusing secondary developments of this software (Seyoum et al., 2012; Son et al., 2016; Zhu et al., 2016). We expected that this application would be difficult to use in our study area due to differences in computer programming. Coupling a model with SWMM for 2D simulation is another way to simulate the spatial distribution of urban inundation (Huong and Pathirana, 2013; Wu et al., 2017).*

*The Integrated Urban Flood Modeling System (IFMS Urban) was developed by the China Institute of Water Resources and*

30  *Hydropower Research (IWHR) in cooperation with other institutions. Based on the simulated results from SWMM, IFMS Urban can simulate the temporal and spatial distribution of urban inundation, and it is compatible with ArcGIS and SWMM. Data conversion and model coupling are accomplished in IFMS Urban, and it does not need additional software programming, which is convenient for researchers and non-expert users."*

35  **Results:**

1. The results for hazard level seem very sensitive to the thresholds chosen. Please give information on the thresholds chosen.

Re: The main basis for the thresholds is according to the relationship between vehicle speed and inundation depth researched in Su et al. (2016). Comparing their results with the study status, we set the three hazard levels for this research. Indeed, different thresholds might inform the results for hazard level and researches on more accurate thresholds are needed in future studies. We

40  will put it in the Limitations and future studies in  the revised manuscript.

*Change in manuscript: The main basis for the choice of thresholds is showm on Page 8, line 25–26, and the additional information is added on Page 12, line 20–22.*

*Page 8, line 25–26 :"based onaccording to the literature (Su et al., 2016) as well and observed data for as actual situation of the study area."*

*Page 12, line 20–22:"Another limitation was that the definition of the thresholds for hazard levels was not considered sufficiently in this study. The results for the three hazard levels would be different if the thresholds changed. Therefore, research on criteria and sensitivity analysis of thresholds is needed in the future."*

2. Results are contradicted. The authors reported on Page 7 line 11-12 "the reduction effects become more evident as hazard level increases", "the roles of LID practices with respect to urban inundation mitigation are less obvious at High levels than thoseat Low levels". So which one is correct?

Re: From line 4-5 on page 7 of our manuscript, our research results show that for the High levels, the **depth reduction** after the construction of LID practices is from 0.11 m to 0.19 m (greater than that for the Low levels) and the **depth reduction rates** are from 22 % to 40 % (lower than those for the Low levels) under Scenarios 1 to 4. We didn't express clearly about the results in our original manuscript but we will improve it in the revised manuscript.

*Change in manuscript: We will mainly talk about the depth reduction rate in section **3.2** and the paragraphs have been improved on Page 8, line 26–Page 9, line 7.*

*"Compared to the benchmark, the ranges of average depth reduction rates were 60–80, 27–54, and 22–40 % at low, medium and high hazard levels , respectively, for Scenarios 1 to 4 (Figure 5). Under different hazard levels, the average depth reduction rates increased from Scenarios 1 to 4. The average depth reduction rates at the low level were 38, 44, 43, and 40 % higher than the high level under Scenarios 1 to 4, respectively. These results suggest that most inundated areas could not be eliminated at the high level because of severe waterlogging."*

3. please show the spatial distribution of reductions in inundation depths instead of average reduction

Re: Figure 5 shows the spatil distribution of reductions in maximum inundation depths of the study area. And from this figure we can see the spatial changes of inundation depth in different scenarios. So we didn't show the spatial distribution of reductions in inundation depths.

*Change in manuscript: Besides, Figure 5 also shows the hazard levels of different scenarios. Therefore, we hope to keep this figure.*

4. please give more information of PP and GR implementation area, otherwise, you cannot say PP performs better than GR

Re: Thanks to point out our careless on the information missing. Data information is as follows:

The available implementation area of PP and GR is 5.95 km$^2$ and 8.92 km$^2$, respectively. The depth reduction rates of 100% PP are 67%, 38% and 23% at Low, Medium and High levels, and the depth reduction rates of 100% GR are 61%, 31% and 21% in three hazard levels. The area reduction rates of 100% PP are 37%, 65% and 67% at Low, Medium and High levels, and the area reduction rates of 100% GR are 32%, 56% and 67% at three hazard levels. Although the implementation area of PP is smaller than GR, the effectiveness of PP on urban immunation mitigation is greater than GR. So we say that PP performs better than GR in this study.

*Change in manuscript: The implementation area of PP and GR is added on Page 7, line 17–18, the reduction data is shown on Figure 5, and the comparison of them are present on Page 10, line 25–28.*

5. in section 3.3 you said the reduction in inundation area under High level was more obvious, but in section 3.1, reduction in inundation depth was less obvious. Please explain.

Re: Poor expression makes this part confusing to be understood but we have improved the expression in the revised manuscript. From the simulated results shown in section 3.2, the ***depth reduction*** after the construction of LID practices is greater but the ***depth reduction rate*** is lower under the High levels compared to Low levels (***question 2, Results***).

In section 3.3, the ***area reduction rate*** is greater under High level compared to other hazard levels (line 22 on page 7). This is because that after the construction of LID practices, in High level, the inundation depth has been decreased and most inundation areas are downgraded from High level to Medium or Low levels, but most inundation areas heavn't been eliminated which make the ***depth reduction rate*** lower than other levels. This is the reason why the ***depth reduction rate*** is lower and the ***area reduction rate*** is greater in High level compared with other levels.

6. please show the spatial distribution of reductions in inundation time instead of average

Re: Through the analysis of inundation depth and inundation area, we can draw the conclusions of this study approximately, and the analysis of inundation time comfirm effectiveness of LID practices from another aspect. Considering from the full text, inundation time is not the key point in this study, so we didn't show the map of inundation time. If necessary we will discuss it further in the revised manuscript.

7. one of the key points in your study is to compare the differences of all scenarios at three hazard levels not to find the differences among three hazard levels

Re: Indeed we both consider the two groups of comparisons in results. From Figure 6 we can see that as the propotion rises from 25% to 100%, the ***depth reduction rate*** (a) and ***area reduction rate*** (b) both increase in the Low, Medium and High levels. It is clear that the reduction rate grows slowly associated with the increasing of proportion of LID implementation from 25% to 100%, which means the efficiency of LID implementation decreases from Scenario 1 to Scenario 4. To better describe the phenomenon, we will built a cost-effectiveness indicator (RPI) in the revised manuscript:

$$\text{RPI} = \frac{R}{P}$$

R means reduction rate of inundation depth and inundation areas, and P means the proportion of LID implementation. From Table 6 we can see that the RPIs of 25% PP+25% GR are always higher than the other scenarios while higher RPI indicates higher efficiency. From the comparisons, we can conclude that the simple increase of the proportion of LID implementation cannot necessarily contribute to the higher efficiency. Finally, we find that the efficiency of 25 % PP + 25 % GR is higher than other scenarios in this study. This indicates that we should not only consider the effectiveness but also the cost of LID practices in the construction of "Sponge City".

Table 6 RPI under differrent scenarios.

| | | 25%PP+25%GR | 50%PP+50%GR | 75%PP+75%GR | 100%PP+100%GR |
|---|---|---|---|---|---|
| Maximum inundation depth | | 0.64 | 0.44 | 0.35 | 0.29 |
| Average inundation depth | Low | 2.40 | 1.48 | 1.05 | 0.80 |
| | Medium | 1.08 | 0.86 | 0.68 | 0.54 |
| | High | 0.88 | 0.60 | 0.48 | 0.40 |
| Average inundation areas | Low | 1.23 | 0.87 | 0.68 | 0.53 |
| | Medium | 2.22 | 1.37 | 0.97 | 0.75 |
| | High | 2.86 | 1.62 | 1.14 | 0.90 |

*Change in manuscript: This has been added on Page 11, line 20–31.*

*"4.3 Cost-effectiveness of LID practices*

*Under Scenarios 1 to 4, the effectiveness of LID practices for urban inundation mitigation increased with more area implementing LID practices. However, Table 4 and Figure 5 show that the reduction rates grew slowly with the increase of LID practices from 25 % to 100 %, which suggests that the efficiency of LID practices decreased from Scenario 1 to Scenario 4. To better describe this phenomenon, we used a cost-effectiveness indicator (CEI) :*

$$CEI = \frac{R}{P} \quad , \tag{1}$$

*where R is the reduction rate of inundation depth and inundation area, and P is the proportion of LID practices. Table 6 shows that the CEI decreased as the proportion of LID practices increased from Scenario 1 to Scenario 4, and the efficiency of the 25 % PP + 25 % GR scenario was higher than other scenarios (even higher than the 100 % PP + 100 % GR scenario). This indicates that simply increasing of the proportion of LID practices is not necessarily more efficient. Therefore, the effectiveness and the cost of LID practices should be considered in the construction of sponge cities."*

**Discussion:**

1) The discussion is lacking depths. What are the same and different points comparing your study and others? What you studied from this research.

Re: Compared to the existing studies about LID, this study tries to explain the cost-effectiveness of LID for urban inundation risk mitigation. Moreover, this study focuses on the cost-effectiveness changes in different hazard levels under different scenarios.

1. The effectiveness of PP for urban inundation mitigation performs better than that of GR in this research. This conclusion might be different in other regions because of the differences of LID parameters, implementation area, spatial pattern, rainfall intensity, rainfall frequency and other factors. But it gives a reference for local residents and policy-maker that PP might be a good choice for local areas because of the great effectiveness and the large potential for reconstruction in the built-up region (PP could be gradually applied in roads and parking lots, while GR is hard to implement in density construction lands, especially in the urban villages);

2. Through the analysis in section 3.2 and 3.3, we can find that in High level, the inundation depth has been decreased and most inundation areas are downgraded from High level to Medium or Low levels, but most high inundation hazard areas heavn't been eliminated and the ***depth reduction rate*** is lower than other levels. This indicates that  LID practices can only ease the inundation depth and downgrade the inundation hazard level in High level. And some other methods of stormwater management should be used together to deal with severe waterlogging in High level areas;

3. Through the analysis in ***question 7, Results***, we find that the RPI decreases as the proportion of LID implementation increases from Scenario 1 to Scenario 4 and the efficiency of 25 % PP + 25 % GR is higher than other scenarios in this study. This indicates that the simple increase of the proportion of LID implementation cannot necessarily contribute to the higher efficiency, and we should not only consider the effectiveness but also the cost of LID practices in the construction of "Sponge City". These findings may provide some suggestions for LID designs in other regions.

*Change in manuscript: The Discussion has been modified and improved on Page 10, line 22.*

2) The discussion on cost-effectiveness completely fell from the sky on page 9 line 3. You neither present how the costs were estimated nor discussed them in the Results.

Re: The main difference among scenarios from Scenario 1 to Scenario 4 is the proportion of LID implementation, and the cost will be higher as the proportion of LID implementation increases. Therefore, we develop a cost-effectiveness indicator (RPI) to discuss on the efficiency of LID practices (***question 7, Results***). We will add these descriptions in the revised manuscript.

*Change in manuscript: This part has been added on Page 11, line 20–31.*

3) In page 9 line 22, "we also find that spatial distribution of landscape patterns ...". This information completely fell from the sky. You neither present them in the Results.

Re: Thanks for pointing out the expression problem that these results are from Kim and Park (2016) and Giacomoni and Joseph (2017), and we will modify it in the revised manuscript.

*Change in manuscript: The sentence has been modified on Page 12, line 30–32.*

*"However, the spatial distribution and landscape patterns of LID practices also contribute to urban flooding mitigation (Giacomoni and Joseph, 2017; Kim and Park, 2016), but few studies have considered these variables."*

4) You reported 25% of PP and GR had the highest efficiency. Is it correct? Do you consider the effect of rainfall intensity and frequency? LID effectiveness is highly related to rainfall intensity and frequency.

Re: You made a very constructive suggestion. We did find that rainfall intensity and frequency will influence effectiveness of LID. However, this study focuses on the trade-offs between implementation cost and effectiveness of LID practices, and we did not change the rainfall intensity or other factors in this study. In our research, once-in-100-years heavy rain happened on 11 May 2014 (144.9 mm) is selected to simulate the urban inundation situation. Because we find heavy rain of this intensity attacks Shenzhen almost very year associated with climate change. In this research, place-based references are provided for the policy-makers, and

we do not suppose all the findings of this research can be directly transferrable to other places, cities even countries but the analytical methods and the efficiency analysis.

*Change in manuscript: Through the comparison between the 25 % PP + 25 % GR scenario and the 100 % PP + 100 % GR scenario, we find that wider implementation of LID practices may not lead to higher efficiency (Page 11, line 27–31). The effect of rainfall intensity and frequency has been added on Page 12, line 22–23.*

*Page 11, line 27–31:"Table 6 shows that the CEI decreased as the proportion of LID practices increased from Scenario 1 to Scenario 4, and the efficiency of the 25 % PP + 25 % GR scenario was higher than other scenarios (even higher than the 100 % PP + 100 % GR scenario). This indicates that simply increasing of the proportion of LID practices is not necessarily more efficient. Therefore, the effectiveness and the cost of LID practices should be considered in the construction of sponge cities."*

*Page 12, line 22–23:"The influences of rainfall intensity and frequency were not considered in this study, which is related to the effectiveness of LID."*

**Specific comments**

1) Page (P) 1 line (L) 15-19, too long to understand.

Re: This study proposes a hydrodynamic inundation model, coupling SWMM (Storm Water Management Model, 1D) and IFMS Urban (Integrated Urban Flood Modeling System, 2D), to assess the effectiveness of LID practices under different scenarios and hazard levels. The results are shown as follows.

*Change in manuscript: The sentence has been simplified on Page 1, line 14–18.*

*"This study used a hydrodynamic inundation model, coupling SWMM (Storm Water Management Model) and IFMS Urban (Integrated Urban Flood Modelling System), to assess the effectiveness of LID under different scenarios and hazard levels."*

2) P1L25, considering cost-effectiveness, you don't give any information on it.

Re: The information about cost-effectiveness is mentioned above (***question 2, Discussion***).

*Change in manuscript: This has been added on Page 11, line 20–31.*

3) P2L4: what are secondary disasters? it is better to delete.

Re: Amended as requested.

*Change in manuscript: This word has been deleted on Page 2, line 7–8.*

4) P3L18-20, there some studies on this topic, please review them.

Re: Amended as requested.

*Change in manuscript: The paragraph has been modified on Page 4, line 1–3.*

*"However, most of these models have a cost, which limits their application, and an open-source model (like Storm Water Management Model, SWMM), with a LID module that can be coupled to simulate urban inundation, is needed (Burns et al., 2015, Hu et al., 2017, Wu et al., 2017)."*

5) P3L29, give rainfall information from April to September.

Re: April to September marks the rainy season in Shenzhen. There are 38 rainstorm days (95% of the whole year) in 2017 and the average rainfall is 170-350 mm every month during this period.

*Change in manuscript: The information has been added on Page 4, line 14–15.*

*"There were 38 rainstorm days (95 % of the year) in 2017 and the average rainfall was 170–350 mm every month during this period."*

6) P4L5-10. simplify the description.

Re: The study site is located in Guangming New District of Shenzhen, China (Fig. 1). The total area of this study site is 37.68 km$^2$ with 69.8 % of it is the construction land. Because of the intensive inundation disasters, Guangming New District was selected as the first pilot area for LID practices in Shenzhen in October 2011. Therefore, there is a need to research the effectiveness of LID practices on urban inundation mitigation in this area.

*Change in manuscript: The paragraphs have been reorganized on Page 4, line 17–20.*

*"The study site was located in Guangming New District of Shenzhen, China, and it is in the Maozhou River Basin (Figure 1). The total area of our study site was 37.68 km2, of which 69.8 % was impervious surfaces. Guangming New District was selected as the first pilot area for LID practices in Shenzhen in October 2011 because of the intensity of its inundation disasters. There is a need to research the effectiveness of LID on urban inundation mitigation in this area. "*

7) P4L12, delete"needed for modeling".

Re: Amended as requested.

*Change in manuscript: The phrase has been deleted on Page 5, line 2.*

8) P4L18-19, how to do.

Re: We resampled the DEM using Resample tool in ArcGIS 10.1.

*Change in manuscript: The sentence has been deleted on Page 5, line 9–10.*

9) P4L20-22, improve.

Re: The reason why we choose the two events is mentioned above (***question 1, Materials and methodology***) and we will improve it in the revised manuscript.

*Change in manuscript: The sentence has been improved on Page 5, line 11–13.*

10) P5L1-3, improve.

Re: We have reorganized section 2.3 and 2.4 in the revised manuscript.

*Change in manuscript: The sentence has been improved on Page 6, line 14–17.*

*"Model coupling occurred in IFMS Urban. First, an unstructured 2D grid model was meshed with an average cell size of 15 m; second, ground elevations were assigned to each grid; finally, each node was linked with a corresponding grid for water exchange, and the distribution of surface inundation was calculated with 2D shallow water equations."*

11) give clear information on the model.

Re: The detailed information about the model is introduced above (***question 5, Materials and methodology***) and we will improve it in the revised manuscript.

*Change in manuscript: The introduction has been added on section **2.3** (Page 5, line 25–Page 6, line 5).*

12) P5L31-32, "we found ..."???

Re: We have not explained the details for this part. In fact, there are some special attributes for buildings on the dense construction land in our research area. Through the detailed urban planning and field investigations of our research area, we found the 80% of the residential lands are urban villages, desnsely constructed on construction lands. The structures and shapes of roofs for urban villages are diversity which makes it difficult to build green roofs on them. More important, the complex owenership and financing pathways which also make it difficult to construct the green roofs for the dense construction lands in our research area. Therefore, we temporarily didn't set green roof in the dense construction land in this study.

*Change in manuscript: The sentence has been improved on Page 7, line 12–14.*

*"Through remote sensing images and field investigations, we found that urban villages have diverse roof structures and shapes, which makes it difficult to implement green roofs."*

13) P6L3, strength? is it density?

Re: We will instead "Construction strength" of "construction density" here.

*Change in manuscript: The phrase has been modified on Page 7, line 19.*

14) P6L19, relative error 30% is acceptable?

Re: Lacking observation data is a universal problem in model simulation, and some models did not have a calibration (Hu et al., 2017). In this study, the relative error of calibration seems a little high, while the relative errors of validation are 5-20%, which is met the requirements of the Standard for Hydrologic Information and Hydrologic Forecasting in China (GBT_22482-2008). If there are more detailed inundation records, the model can be further improved in the future study. We will discuss the limitation in section 4.4 Limitations and future studies.

*Change in manuscript: The phrase has been modified on Page 8, line 14–16 and Page 12, line 19–20.*

*Page 8, line 14–16:"In this study, the relative error of calibration were a little higher, while the relative errors of validation were 5–20 %, which met the requirements of the Standard for Hydrologic Information and Hydrologic Forecasting in China (GBT_22482-2008)."*

*Page 12, line 19–20 :"Moreover, the accuracy of the coupled model could be further increased with more observed data and information."*

15) P6L24-25, give more literature to support

Re: Amended as requested.

*Change in manuscript: The standard has been added on Page 8, line 14–16.*

25    Therefore, the goal of this study is to demonstrate through a case study the effectiveness of LID practices to mitigate urban

inundation in an urban watershed. The specific objectives were to (1) establish a 1D-2D hydrodynamic model coupled SWMM

and IFMS Urban; and (2) evaluate the effectiveness of LID practices under different scenarios and hazard levels; and (3) explore

the effiency of designed scenarios that related to the effectiveness of LID practices and the proportion of implementation areas.

This study hopes to enrich the inundation mitigation research of LID on an urban watershed scale and provide some references to

30    urban stormwater management and inundation mitigation for local government.

*Change in manuscript: The paragraphs has been modified on Page 3, line 31–Page 4, line 8.*

*"Peak flows reduction, runoff reduction, and hydrograph delays are widely used indexes for evaluating the performance of LID*

*practices (Ahiablame and Shakya, 2016; Qin et al., 2013; Zhang et al., 2016). However, these indexes are not intuitive, and the*

*performance of LID practices for urban inundation is more useful for local residents, such as providing a guide for their travel*

35    *behaviour. Some 1D-2D models have been applied for flood management, such as ESTRY-TUFLOW (Fewtrell et al., 2011),*

*InfoWorks ICM (Russo et al., 2015) and MIKE FLOOD (Loewe et al., 2017). However, most of these models have a cost, which*

*limits their application, and an open-source model (like Storm Water Management Model, SWMM), with a LID module that can*

*be coupled to simulate urban inundation, is needed (Burns et al., 2015, Hu et al., 2017, Wu et al., 2017).*

*The goal of this study was to evaluate the effectiveness of LID practices to mitigate urban inundation in an urban watershed using*

40    *a case study. The specific objectives were to establish a 1D-2D hydrodynamic model that coupled SWMM and IFMS Urban,*

*evaluate the effectiveness of LID practices under different scenarios and hazard levels, and explore the efficiency of the LID scenarios. We intended this study to enrich LID inundation mitigation research at the urban watershed scale and to provide a reference for urban stormwater management and inundation mitigation for local governments."*

2. 2.2 Data, part 4: What were the criteria for removing nodes and pipelines? A reduction from 4502 to 597 pipelines and from 1175 to 653 nodes seems a bit more that deleting some redundant and incorrect data. How was it tested that the data were redundant?

Re: Indeed, the actural drainage networks are compulsory and substantial. Nontheless, SWMM cannot accurately simulate when the data is huge. Besides, after the data conversion process for applying into the SWMM, some overlaps and break points for the pipelines are generated, which makes lots of nodes and pipelines useless. Therefore, we have to simply the drainage data for model building and the criteria shown below:

a. Add nodes when the pipeline is too long;

b. Keep or add the corner nodeschanging diameter nodes, or large variation range of slope nodes;

c. Keep the parallel pipelines and nodes on both sides of the roads;

d. Delete the useless nodes and pipelines in this model.

*Change in manuscript: The criteria have been added on Page 5 line 16–21 .*

*"We simplified the drainage data for building the model because the urban pipe network is intricate and substantial: add nodes when the pipeline is too long; keep or add the nodes that change the diameter and slope of pipeline; keep the parallel pipelines and nodes on both sides of the roads; and delete the useless nodes and pipelines in this model. Finally, the 4502 pipelines and 1175 nodes in this study were generalized to 597 pipelines and 653 nodes, including 56 outlets and 597 inspection nodes (Figure 2)."*

3. 2.4 Coupling the SWMM/IFMS Urban models: As written above, I think that one does not learn much about the coupling. Also, Figure 4 does not help in this respect. One just learns that the models were coupled. But how were they coupled? Is inflow and outflow from and to manholes possible? What were the criteria for inflow and outflow? What was the spatial resolution of the geometry of a street? What timesteps were chosen for coupling? Either more discussion about the coupling is needed, which means that one also need to know more about the numerical schemes used for the two different models, or it does not make sense to have a section for this part.

Re: SWMM is a 1D rainfall-runoff model which use the given hydrology data and hydrodynamics to simulate the quantity and quality of rainfall-runoff. Notheless, when the node overflow occurs, SWMM can't simulate the spatial and temporal distributions of surface inundation, but the IFMS Urban can using 2D shallow water equations. However, the simulations of IFMS Urban must base on the simulated results of SWMM. So we coupled these two models to realize the simulate on the spatial and temporal distributions of surface inundation. What's more, the process of data conversion and model coupling are all accomplished in IFMS Urban, and it doesn't need other software programming or specialized knowledge, which is convenient for researchers and non-expert users. So we don't want to make it complicated or list algorithm and formula in this part. The spatial resolution of the geometry of a street is 15 m. The timestep of calculation is 10 s and the timestep of output is 200 s.

*Change in manuscript: Here we listed the main processes of model coupling for readers to understand the important parts in the coupled model (Page 6, line 14–17). The principles of calculation engine and details can be found on the useral manual, while we did not want to make it complex in this part of study. If necessary, we could add the introduction of algorithms and formulas in the next vision.*

*"Model coupling occurred in IFMS Urban. First, an unstructured 2D grid model was meshed with an average cell size of 15 m; second, ground elevations were assigned to each grid; finally, each node was linked with a corresponding grid for water exchange, and the distribution of surface inundation was calculated with 2D shallow water equations."*

4. Page 5, lines 18-20: This sentence is unclear. Also: What is innovative about the coupling?

Re: SWMM is a 1D hydrodynamics model which can simulate the quantity and quality of rainfall-runoff but it can't simulate the urban inundation, while the IFMS Urban is a 2D model which can simunate the urban inundation but it must base on the results of SWMM. Through coupling, we build a 1D-2D hydrodynamic model that can simulate the spatial and temporal distributions of surface inundation. Based on this coupled model, we can evaluate the effectiveness of LID from inundation depth, inundation area and inundation time. And this coupled model both takes in the advantages of SWMM and IFMS Urban (open-source, free, great compatibility with ArcGIS and 2D inundation simulation), which is convenient for researchers and non-expert users.

*Change in manuscript: The sentence has been removed and the reasons why we choose and couply the two models have been shown on Page 5, line 25–Page 6, line 5.*

*"SWMM is an open-source model that can simulate dynamic runoff quantity and quality from urban areas, and it has been widely used to simulate the hydrologic performance of LID practices (Rossman, 2010; Wu et al., 2013). However, SWMM cannot simulate the spatial and temporal distributions of surface inundation. Recently, some scholars have conducted simulationsusing secondary developments of this software (Seyoum et al., 2012; Son et al., 2016; Zhu et al., 2016). We expected that this application would be difficult to use in our study area due to differences in computer programming. Coupling a model with SWMM for 2D simulation is another way to simulate the spatial distribution of urban inundation (Huong and Pathirana, 2013; Wu et al., 2017).*

*The Integrated Urban Flood Modeling System (IFMS Urban) was developed by the China Institute of Water Resources and Hydropower Research (IWHR) in cooperation with other institutions. Based on the simulated results from SWMM, IFMS Urban can simulate the temporal and spatial distribution of urban inundation, and it is compatible with ArcGIS and SWMM. Data conversion and model coupling are accomplished in IFMS Urban, and it does not need additional software programming, which is convenient for researchers and non-expert users."*

5. Page 5, line 26: Why was a geostatistical method (Kriging) used for interpolation? I do not see the connection to geostatistics for a digital elevation model in a city.

Re: We need DEM when building the 2D model. However, the accuracy of DEM production from Geospatial Data Cloud can not meet our demand (for example, 6 m, 13 m). However, the high accuracy DEM is confidential and difficult to obtain in China. Alternatively, we find the ground elevation of nodes in pipe network data has a higher accuracy (for example, 6.588 m, 13.483 m), and the nodes on the roads are relatively dense. So we use a geostatistical method (Kriging) to get a high accuracy DEM of the roads with the elevation data of nodes on the roads.

*Change in manuscript: The additional information is added on Page 12, line 16–19.*

*"Lacking accurate data is a common limitation for most studies. In this study, highly accurate elevation data for the study area is confidential and difficult to obtain; therefore, the ground elevation of streets were interpolated from the dense nodes of the pipe network. This method may have affected the simulation results."*

6. Page6, top: Please explain why green roofs should not be possible in a dense construction land.

Re: We have not explained the details for this part and thank you for your kind reminder. In fact, there are some special attributes for buildings on the dense construction land in our research area. Through the detailed urban planning and field investigations of

our research area, we found the 80% of the residential lands are urban villages, desnsely constructed on construction lands. The structures and shapes of roofs for urban villages are diversity which makes it difficult to build green roofs on them. Thereforce, we temporarily didn't set green roof in the dense construction land in this study.

*Change in manuscript: The explanation  is added on Page 7, line 12–14.*

*"Through remote sensing images and field investigations, we found that urban villages have diverse roof structures and shapes, which makes it difficult to implement green roofs."*

7. Modeling part (Section 2): Please explain how green roofs and permeable pavements are realized in the model. I assume that a storage for a roof area is assigned (or an existing one is increased) and that there is a soil compartment which gets a connection to the paved area if the pavement is permeable. As this is the key process that is here investigated, I think it is necessary to outline these things (and it is not enough to refer to the manuals of the models).

Re: The simulation designs and parameter setting for PP and GR are listed in Table 1 of our paper, which are strictly desinged according to the manual of SWMM and some highly cited studies of LIDs  (Ahiablame and Shakya, 2016; Chui et al., 2016; Kong et al., 2017; Qin et al., 2013).

*Change in manuscript: We modified the paragraphs on Page 7, line 11–12.*

*"The parameters for PP and GR are listed in Table 1, which were designed based on SWMM requirements and LID research (Ahiablame and Shakya, 2016; Chui et al., 2016; Kong et al., 2017; Qin et al., 2013)."*

8. Page 7, top: Please explain why the classification in hazard levels is made. What can be learned from the classification? It is written that the changes of inundation level are different for the different classes. But what does one make out of this fact? More discussion about consequences would be useful.

Re: Through the classification in hazard levels, we can explore the effectiveness of LID practices in different hazard levels, especial in the High level. Through the analysis in section 3.2 and 3.3, we can find that in the High levels, the inundation depth has been decreased (***depth reduction rates*** are from 22% to 40 %) and most inundation areas are downgraded from High levels to Medium or Low levels (***area reduction rates*** are from 71% to 90 %), but most inundation areas heavn't been eliminated and the ***depth reduction rate*** is lower than other levels (lower 38-40% than Low level). This indicates that LID practices can only ease the inundation depth and downgrade the inundation hazard level and can't thoroughly resolve the inundation problem in High level. And some other methods of stormwater management should be used together to deal with severe waterlogging at High level areas.

*Change in manuscript: The discussion about effectiveness under hazard levels has been added  on Page 11, line 10–19.*

*"4.2 Effectiveness at different hazard levels*

*At the high level, the average depth reduction rates decreased from 22 % to 40 %, and the average area reduction rates decreased from 71 % to 90 % under Scenarios 1 to 4. These results showed that the inundation hazard eased at a high level with the implementation of LID practices. However, at the high level, the average depth reduction rates were still 38–40 % lower and the average inundation time was 2.5–5.9 h longer when comperaed to the low level; this indicates that LID practices are more effective for urban inundation mitigation at a low hazard level. The hazard level analysis showed that although LID practices can downgrade the inundation hazard level to medium or low, most inundated areas cannot be eliminated at a high hazard level. This means that the inundation problem could not been resolved only with LID practices; other stormwater management methods should be applied to manage severe waterlogging in high hazard areas, such as restoring river systems, establishing urban wetlands, and improving urban drainage infrastructure."*

9. Page 7, line 4: Please name scenarios 1 to 4

Re: Amended as requested.

*Change in manuscript: We modified the scentence on Page 8, line 29.*

5   10. Figure 6: What is meant by percentage GR and PP? Both with the same percentage?

Re: The proportion means the percentage of the total available implementation areas of LID. Here the **percentage GR and PP** means the proportion of Scenario 1 to Scenario 4 (from 25% to 100%) in Figure 6.

*Change in manuscript: We have modified this figure to show the data of scenario 1 to scenario 6 (Figure 5).*

10   11. Page 7, lines 14-18: I do not see where this conclusion comes from. Is this concluded from the numbers in Table 4? What is here meant by performance? Reduction of maximum inundation? This paragraph needs clarification.

Re: We did not put the data in the part that the **depth reduction rates** of 100% PP are 67%, 38% and 23% at Low, Medium and High levels, and the **depth reduction rates** of 100% GR are 61%, 31% and 21% at three hazard levels. Here the performance means the average depth reduction rate. We will reorganize this paragraph.

15   *Change in manuscript: The data has been added in Figure 5, and the paragraph has been reorganized on Page 9, line 8–15.*
*"Figure 5 shows that the average depth reduction rates of 100 % PP and 100 % GR scenarios were between the 25 % GR + 25 % PP and 50 % GR + 50 % PP scenarios under different hazard levels. These results suggest that LID combinations may be more effective in reducing urban inundation than a single type of LID practice. Based on the comparison of the two LID practices, we found that the average depth reduction rates of the 100 % PP scenario were 67, 38 and 23 % at the low, medium and high levels,*
20   *respectively. These were 6, 7, and 2 % higher than the average depth reduction rates of the 100 % GR scenario.These results suggest that PP may perform better than GR for reducing the depth of inundation."*

12. Page 7, lines 28-31: Again it is not clear where these numbers come from. I do not find it in the Figures. In Figure 6, the single 100 percent cases are not shown.

25   Re: We did not put the data in the part that the **area reduction rates** of 100% PP are 37%, 65% and 67% at Low, Medium and High levels, and the **area reduction rates** of 100% GR are 32%, 56% and 67% at three hazard levels. We will add the data in Figure 6.

*Change in manuscript: The data has been added in Figure 5.*

13. Page 8, line 11: This needs explanation. Why is it difficult to mitigate? Is the reason the topography? I think that
30   such a statement needs to be more specific.

Re: The topographical attributes, such as concaves and potholes, are easy to lead to some places got inundation on the road surfaces. If these places are not or not enough drainage pipes to drainage the rainwater, it is difficult for them to mitigate the influences of urban inundation even there are LIDs. Because of these long-time inundation time areas, the average inundation time increases 0.1 h after the implementation of LID practices (**question 15**).

35   *Change in manuscript: We modified the sentences and mainly explained why the average inundation time increases after the implementation of LID practices. on Page 10, line 10–21.*
*"This result did not indicate that LID practices cannot decrease inundation time or that the model had errors. The inundation time decreased for all hazard levels, butfor the low and medium levels, some areas inundated for a short-time were no longer flooded, which resulted in a different urban inundation area after the implementation of LID practices. Therefore, the average inundation*

*time was longer than before LID practices were implemented at the low and medium levels. As LID practices were implemented, the average inundation time decreased continuously from 4.1 to 2.3 h under scenarios 1 to 4."*

14. Page 8, lines 12-13: How can one see in these figures that the infrastructure is not perfect? And what infrastructure is here meant and how does it influence the inundation? Also: How can one see from these figures that the LID practices are not perfect? In which sense are they not perfect?

Re: Here we want to explain why some places are difficult to mitigate (***question 13***). These sentences are not rigorous and we will modify them in the revised manuscript.

*Change in manuscript: This scentence has been removed.*

15. Page 8, lines 13-15: I could not follow the reasoning. Why does the mitigation of short-time inundation areas lead to an increase in the average inundation time with LID measures? Is here something meant along the lines: If a storage due to green roofs helps to keep water back, leading to less inundation depth, the storage will at the same time lead to a longer inundation time (it holds the water back, but releases it eventually)? I am just guessing and I think this needs a
better explanation.

Re: Indeed, this is because the statistical number of urban inundation areas are not the same before and after the implementation of LID practices. Here we want to explain why the average inundation time increases 0.1 h after the implementation of LID practices. Because of the implementation of LID practices, the inundation time has been decreased in all hazard levels. However, for the Low level some short-time inundation areas previously affected by surface runoff are freed from urban flooding after the construction of the LID projects, which makes the total number of inundation areas decreases after the implementation of LID. More important, the most freed areas are short-time inundation areas. Although LID practices make existing urban inundation areas' inundation time shorten, the statistical data suggest that the average of the lasting inundation areas' inundation duration is a little longer than that before LID practices. It is also suggests the great effectiveness of LID practices at Low level. We will modify the sentences in line 11-15 and make them clearer to understand.

*Change in manuscript: Same to Question 13.*

16. Section 4.1, Comparison of permeable pavement and green roofs: What is the reasoning of the different effects? This should be explained based on the mitigation mechanisms. The last sentence sounds a bit strong. I do not think that one test case can use as a proof, if no general reasoning is given for the different performances.

Re: The available implementation area of PP and GR is 5.95 $km^2$ and 8.92 $km^2$, respectively. Although the implementation area of PP is smaller than GR, the effectiveness of PP on urban immunation mitigation is greater than GR in this study (***question 11, 12***). Except the differences of LID parameters, the reason of the different effects might be that PP is built both on low ad high construction lands, while GR is only built on low density construction lands. Indeed, the effectiveness of PP for urban inundation mitigation were different from studies (Qin et al., 2013, Ahiablame and Shakya, 2016, Zhang et al., 2016, Hu et al., 2017), and PP can not always perform better because that the effectiveness is depended on the parameters, implementation area, spatial pattern, rainfall intensity, rainfall frequency and other factors in different regions. Here we want to give a reference for local government that PP might be a good choice for local areas because of the great effectiveness and the large potential for reconstruction in the built-up region (PP could be gradually applied in roads and parking lots, while GR is hard to implement in density construction lands, especially in the urban villages).

*Change in manuscript: The paragraph has been added on Page 10, line 25–Page 11, line 3.*

*"Our analysis showed that, although the implementation area of PP was less than GR, PP provided better urban inundation mitigation than GR. This result may have been due to differences in the LID parameters, but it may also have been caused by the PP's more diffuse spatial pattern. PP have shown varying effectiveness for urban inundation mitigation in different studies (Ahiablame and Shakya, 2016; Hu et al., 2017; Qin et al., 2013; Zhang et al., 2016), and PP cannot always perform better because the effectiveness depends on the characteristics, implementation area, spatial pattern, rainfall intensity and rainfall frequency in different regions. Our study shows that PP may be a good choice for local governments because of its effectiveness for stormwater management and its potential use for reconstruction in built-up areas. PP could be gradually applied to roads and parking lots, while GR is harder to implement in densely urbanized areas, especially in the urban villages. "*

17. Page 8, line 29: I would be a bit more careful with the word 'comprehensively'. The paper shows one case study. I do not think that this is a comprehensive exploration of inundation mitigation in an urban watershed.

Re: We will delete the word.

*Change in manuscript: The word has been deleted on Page 11, line 23.*

18. Page 9, lines 10-14: As before, I do not see the point about infrastructure. How is poor infrastructure reflected in the model? If not at all: How can one draw any conclusions about this point from a modeling study that does not capture this effect? If yes: What exactly is meant by poor infrastructure and how is this realized in the model?

Re: The scentences in lines 10-14 are not rigorous. Indeed, we find that the efficiency decreases as the proportion of LID implementation increases from Scenario 1 to Scenario 4 and the efficiency of 25 % PP + 25 % GR is higher than other scenarios in this study. This indicates that the greater proportion of LID implementation might not lead to the higher efficiency, and we should not only consider the effectiveness but also the cost of LID practices in the construction of "Sponge City".

*Change in manuscript: The sentence has been deleted on Page 12, line 10–13 and the paragraph has been reorganized in section 4.3.*

19. Page 22-23: Maybe this sentence is only not formulated well. But I do not see how from this study one could see anything about landscape patterns ('we find that the...' sounds as if it is a conclusion from this study). The landscape patterns are not discussed, so one cannot conclude about this point. For this reason, I can also not see how 'this provides a new perspective'. Or is here simply meant that this point should be studied in the future? In this case the sentences need to be reformulated.

Re: Thanks for pointing out the expression problem that these results are from Kim and Park (2016) and Giacomoni and Joseph (2017), and we will modify it in the revised manuscript.

*Change in manuscript: The sentence has been modified on Page 12, line 30–32.*

*"However, the spatial distribution and landscape patterns of LID practices also contribute to urban flooding mitigation (Giacomoni and Joseph, 2017; Kim and Park, 2016), but few studies have considered these variables"*

20. Conclusions: I think it should be mentioned that the findings in this study apply to the one test case considered. It is not clear if the results are more general and could be transferred to other sites. In particular: Numbers can certainly not be transferred.

Re: This study is a simulation-based research on a local basis. Although the results cannot be transferable to other places directly, the analytical methods, including the coupling model, cost-effectiveness analysis during the sponge city construction can be transferable. We will list the main conclusions below:

1. The coupling model with SWMM and IFMS Urban can be applied to evaluate the effectiveness of LID for urban inundation risk mitigation and can be transferred to other sites.

2. The effectiveness of PP for urban inundation mitigation performs better than that of GR in this research. This conclusion might be different in other regions but it gives a reference for policy-maker on a local basis.

3. LID practices can only ease the inundation depth and downgrade the inundation hazard level but can't thoroughly resolve the inundation problem in High level. Therefore, some other methods of stormwater management should be used together to deal with severe waterlogging at High level areas.

4. The greater proportion of LID implementation might not lead to the higher efficiency, and we should not only consider the effectiveness but also the cost of LID practices in the construction of "Sponge City".

*Change in manuscript: The conslusion has been improved on Page 13, line 4–12.*

[revised manuscript text omitted]

**A list of modifications related to comments**

Please notice that page and line numbers are those of the revised version. And because of the deletion of Figure 4, the Figure 5 is the old Figure 6 and the Figure 4 is the old Figure 5.

[revised manuscript text omitted]

"—" means data miss, "RE" means "relative error", unit: m.

**Table 4: Maximum inundation depth under different scenarios.**

| | Bench mark | 100 % PP | 100 % GR | 25 % PP+25 % GR | 50 % PP+50 % GR | 75 % PP+75 % GR | 100 % PP+100 % GR |
|---|---|---|---|---|---|---|---|
| maximum inundation depth (m) | 0.69 | 0.59 | 0.59 | 0.58 | 0.54 | 0.51 | 0.49 |
| Reduction rate (%) | - | 14 | 14 | 16 | 22 | 26 | 29 |

**Table 5: Inundation time under different scenarios and hazard levels.**

| | Bbenchmark | 100 % PP | 100 % GR | 25 % PP+25 % GR | 50 % PP+50 % GR | 75 % PP+75 % GR | 100 % PP+100 % GR |
|---|---|---|---|---|---|---|---|
| Low (h) | 3.4 | 3.3 | 3.3 | 3.7 | 3.3 | 2.5 | 2.2 |
| Medium (h) | 7.7 | 7.5 | 7.7 | 8.2 | 7.1 | 6 | 4.7 |
| High (h) | 10.6 | 9.3 | 8.4 | 9.6 | 7.6 | 6 | 4.7 |
| Total (h) | 4 | 3.6 | 3.6 | 4.1 | 3.6 | 2.8 | 2.3 |

**Table 6: CEI under different scenarios.**

| | | 25 % PP+25 % GR | 50 % PP+50 % GR | 75 % PP+75 % GR | 100 % PP+100 % GR |
|---|---|---|---|---|---|
| Maximum inundation depth | | 0.64 | 0.44 | 0.35 | 0.29 |
| Average inundation depth | Low | 2.40 | 1.48 | 1.05 | 0.80 |
| | Medium | 1.08 | 0.86 | 0.68 | 0.54 |
| | High | 0.88 | 0.60 | 0.48 | 0.40 |
| Average inundation areas | Low | 1.23 | 0.87 | 0.68 | 0.53 |
| | Medium | 2.22 | 1.37 | 0.97 | 0.75 |
| | High | 2.86 | 1.62 | 1.14 | 0.90 |

---

## Referee Report (RR1)

Review: Revisions of 'Effectiveness of low impact development for urban inundation risk mitigation under different scenarios: a case study in Shenzhen, China'

J. Wu, R. Yang and J. Song

The manuscript has been revised and the authors have replied to all my comments. To some extent, the comments have been taken into account. The manuscript no longer puts the focus on the coupled model itself, which I think is good (as outlined in the previous review: there are several such models). Also, some more explanation about the setup of the model and the effects related to risk levels is given. However, I find the discussion of the results is still superficial. There is no explanation about mechanisms that cause the findings of the paper (impact of LID measures). Also, there is now a new measure (pointed out at one point as a focus of the paper), which is not explained and not discussed. I still think that due to that, the scientific content of the paper is rather lean. It is pointed out in the manuscript that the effect of LID measures is large if some measures are taken but not increased much further if more measures are taken. One could consider that as a finding that is worth publishing. However, one does not learn about the reasons behind this and so cannot really judge. With more discussion one could certainly learn more about the effect of LID measures. I think that revisions are needed for publication of the paper.

- Page 1, line 25: 'Urban-Rural' instead of 'Urban-Rrural'.

- Page 3: I the revised manuscript, the focus is no longer on the coupled model, which I appreciate. However, it is written on page 3 at the end of Section 1 that the reason to present the coupled model used here is that other models (and some are listed) cost money. For this reason, open source models are needed. I completely agree to that and if an open source solutions would be presented this would be a very positive aspect. However, I did not find anything about IFMS as an open source code. SWMM is of course well known and it is easy to find. With IFMS I was not successful. If is it an open source model, it would be good to give reference to where it could be found or obtained. If it is not an open source model, the claim should not be made, or rather, it would then not make sense to write much about the need of open source models, as this is not answered in this paper.

- Page 4, line 8: I think the formulation 'useless nodes and pipelines' is chosen not so well. What is a useless node or pipeline? In reality each pipe was built for a purpose, so in principle no part of the pipe network is useless.

- I would suggest to merge Sections 2.3 and 2.4, as both just describe the model that was applied. The algorithm how water is exchanged between the surface and the pipe network is still not clear to me. Maybe it is not so crucial for the paper, as its focus is not any more on the model.

- Section 2.5: I think it would be good to give some explanation why these two measures were chosen. Also, it is not very clear to me what is meant by the percentages.

It is clear that it means that a certain percentage of all possible GR measures (for example) is considered. But at which locations were the roofs chosen that lead to a certain percentage of all possible measures? I am quite sure that it matters where they would be located (effective close to hot spots, less effective further away). Were they distributed equally over the domain?

- Page 6, line 12: How can one conclude that the cause of the high level is due to severe waterlogging? It might be the cause in reality, but the model will only reproduce causes that are in the model. How were waterlogging conditions implemented in the model?

- Section 4.1: I find it still a problem that the influence of storage is not discussed at all. The effects discussed in the manuscript are certainly caused by the storage assigned to the LID measures. Green roofs have a certain volume. If it is full, no more water can be held back. With more volume, more water could be held back. Permeable pavement gives access to the soil, which is a very large storage. But there is an infiltration rate, so that the storage can only be filled with a certain speed. I think a lot of the effects can be explained with this aspect and it should be discussed.

- The finding that 25 percent of both measures are more efficient than 100 percent of one measure can also be explained with the storages assigned to the LID measures, I assume. A certain storage volume is needed to hold a certain flood volume back. So if measures are taken that provide this volume, the measure is fully efficient. If more is provided, it does not add to further mitigation. In the paper one gets the impression that the fact that less percentage of several measures is more efficient than more percentage of one measure is an unexplainable fact that is here 'found'. In general, the results should be discussed more to get to general conclusions.

- Section 4.3: The cost-effectiveness indication is newly introduced and was not part of the first manuscript. I am not convinced by this indicator. It has a unit that depends on the analyzed quantity and the range would go to infinity if no measures are taken. It is a very non-linear function of the percentage of the measures. This makes it difficult to interpret. If it should be used, it needs to be described very clearly what a certain number of the CEI means. The index comes out of the blue and it is nowhere discussed why this is a good index for the effectiveness of a LID measure.

- Page 9, lines 4-5: I do not see how the conclusion can be made that the model can be used for different cities and countries. In principle, of course any model could be used, but if the meaning is just that, the sentence is trivial. I would delete this sentence.

---

## Author Response (AR2)

**RESPONSE TO THE EDITOR**

Dear Authors,

Your revision has improved the manuscript; however, there is still significant room for further improvement. The reviewers provide very helpful and detailed suggestions. Particularly, some more analyses to test the sensitivity of indexes and discussion about your results are necessary.

I ask you to revise your manuscript in accordance with all the comments and recommendations of each of the reviewers. When you have completed your revision, please submit your revised manuscript with the changes marked, and a detailed item-by-item response to each of the reviewer's comments.

Best regards

Heidi Kreibich

Dear Editor,

We would like to thank the anonymous referees for his/her comments that have been useful to improve some aspects of our manuscript. We have major revised our manuscript and a summary of the revision is provided as the following. The sensitivity analysis has been added in this revision and the discussion about the results on **Section 4.1 and 4.3** has been promoted. Other details of the changes are presented in the revised manuscript. We deeply appreciate your consideration of our work. Please do not hesitate to contact us for any queries.

Best regards

On behalf of all the authors

Yang Rui

**RESPONSE TO THE REFEREE #1**

Dear Referee #1:

Thank you for the valuable comments. Lacking accurate data is one of the main reasons for the relative error, and after running the

5    5 % and 10 % scenarios, we further verify the conclusion of this paper. Our responses to the questions are listed below, and we greatly appreciate your time and efforts to help us to improve our manuscript for further revision and publication.

Best regards

On behalf of all the authors

10    Yang Rui

**suggestion: resample in ArcGIS for upscaling is commonly used, but for downscaling, it requires more discussion.**

Re: Thank you for your suggestion and we didn't resample the DEM in this manuscript.

*Change in manuscript: no change.*

**Question1: P5L24, relative error 0-30%. The reference listed is 5-20%, smaller than 30%. Please give more discussion. What is the impact of this relative error on your conclusions?**

Re: In this study, the relative error of calibration was 0-30%. This may because some measures of protecting flood were not considered in this modle, such as drainage pump station, river channel et. al. Besides, the low accuracy of observed data also

20    directly affected the accuracy of model. Indeed, lacking data is a common limitation for most studies. And some models didn't get calibrated (Hu, 2017). To further verified the applicability of the model, we chose other rainfall (10 May 2016) for validation. And the relative errors of validation were 5–20 %, which met the requirements of the Standard for Hydrologic Information and Hydrologic Forecasting in China (GBT_22482-2008). After considering the available data and the results of calibration and validation, we think the accuracy of the model is acceptable. Relative discussion was added at Section 4.4.

25    *Change in manuscript: We modified the sentences on:*

*Page 6 line 1-2:"To further confirm the applicability of the model, the rainfall and inundation data on 10 May 2016 was chosen to validate the coupled model."*

*Page 9 line 14-16:"Moreover, the accuracy of the coupled model could be further increased with more accurate observed data and information of infrastructure, such as drainage pump station and river channel."*

**Question2: Is CEI a good index for cost-effectiveness analysis? in your study, the scenario of 25% has the best performance followed by 50%,75%, and 100%. How about 5% and 10%, could you run for 5% and 10% and compare them? If 5% and 10% have higher CEI than 25%, CEI may not be a suitable index. I also suggest trying the reduction values not reduction ratio for CEI to see the results.**

35    Re: In the construction of Sponge City, people paid more attention to the effectiveness,while ignored the cost of LID. If we pursued the best effectiveness, we would not support such huge cost. Therefore, we want to find wheather there is a best efficiency scenario considering both the effectiveness and the cost. The cost-effectiveness indicator has been applied on Wu et al. (2017), although CEI might not be the best index for cost-effectiveness analysis, considering the available data and reasonable assumption, it is acceptable in this paper. The results of reduction values have the same performance with the reduction ratio.

After adding two scenarios, 5 % GR + 5 % PP and 10 % GR + 10 % PP, we further verified the conclusion that: wider implementation of LID practices may not lead to higher efficiency, and there exist a highest efficiency scenario (10 % GR + 10 % PP in this paper). Therefore, we should not only consider the effectiveness but also the cost during the construction of Sponge City.

*Change in manuscript: Two scenarios has been added on Page 5 line 17-24, and Figure 4, Figure 5, Table 4, Table 5 and Table 7 has been updated. And more details has been discussed on Section 4.3 (Page 8 line 26-Page 9 line 10).*

*"And we can clearly find that the reduction rates of maximum inundation depth are 7, 16, 22, 26 and 29 % from scenario 2-6 and the CEI has reduced continuously, especially from scenario 4-6. This indicates that wider implementation of LID practices may not lead to higher efficiency.*

*One of the causes behind the phenomenon is that LID practices can not control all the runoff of the watershed. Indeed, the runoff might not only come from sub-catchments around the inundation areas, but also come from other sub-catchments through the roads and pipe networks. And in this study, there are still some areas that can not implement LID practices. Therefore, the runoff from these areas can not be controlled by LID practices and directlly influenced the effectiveness of inundation mitigation.*

*The phenomenon is common. In urban watershed, we could not transform all the roofs and roads to LID practices, and there are still some impervious covers that could influence the inundation that LID practices can not control. Therefore, we should recognize the insufficients of LID practices, and consider combine other measures such as restoring river systems, establishing urban wetlands, and improving urban drainage infrastructure to further promote the effectiveness on inundation mitigation. Besides, properly implementing construction intensity of LID practices to achieve optimal efficiency in urban watershed will be very important for the construction of Sponge City."*

**References**

Hu, M., Sayama, T., Zhang, X., Tanaka, K., Takara, K., and Yang, H.: Evaluation of low impact development approach for mitigating flood inundation at a watershed scale in China, J Environ Manage, 193, 430-438,

Wu, X., Wang, Z., Guo, S., Liao, W., Zeng, Z., and Chen, X.: Scenario-based projections of future urban inundation within a coupled hydrodynamic model framework: A case study in Dongguan City, China, Journal of Hydrology, 547, 428-442, 10.1016/j.jhydrol.2017.02.020, 2017. 10.1016/j.jenvman.2017.02.020, 2017.

Dear Referee #2:

Thank you for the positive comments and constructive suggestions on this paper. The discussion about the effects of parameters

5     has been promoted on **Section 4.1**, and the explanation about mechanisms behind the results has been discussed on **Section 4.3**. From this paper we can find the insufficients of LID practices in severe waterlogging, and some other measures such as restoring river systems, establishing urban wetlands, and improving urban drainage infrastructure are pointed out just as suggestions to solve severe waterlogging together with LID practices.

Our responses to the questions are listed below, and we greatly appreciate your time and efforts to help us to improve our

10     manuscript for further revision and publication.

Best regards

On behalf of all the authors

Yang Rui

1. Page 1, line 25: 'Urban-Rural' instead of 'Urban-Rrural'.

Re: The error has been modified.

*Change in manuscript: The error has been modified on Page 1, line 27.*

2. Page 3: I the revised manuscript, the focus is no longer on the coupled model, which I appreciate. However, it is written on page 3 at the end of Section 1 that the reason to present the coupled model used here is that other models (and some are listed) cost money. For this reason, open source models are needed. I completely agree to that and if an open source solutions would be presented this would be a very positive aspect. However, I did not find anything about IFMS as an open source code. SWMM is

25     of course well known and it is easy to find. With IFMS I was not successful. If is it an open source model, it would be good to give reference to where it could be found or obtained. If it is not an open source model, the claim should not be made, or rather, it would then not make sense to write much about the need of open source models, as this is not answered in this paper.

Re: Originally, we aimed to express that, as an open source and free model, SWMM has been coupled with other models, such as BreZo (Burns et al., 2015) and LISFLOOD-FP (Wu et al., 2017), to simulate urban inundation in these years, which means that

30     the coupled models based on SWMM is needed in future research. Therefore, like other studies, we establish a 1D-2D hydrodynamic model that coupled SWMM and another model, IFMS Urban. Unfortunately, due to the inappropriate expression, IFMS Urban was mistaken as an open source model. Indeed, IFMS Urban is a commercial model. Due to the most modules of it are free for research, we chosed IFMS Urban in this study. In order to eliminate the misunderstanding, this sentence was revised in the manuscript.

35     *Change in manuscript: The sentence has been modified on Page 3 line 9–13 .*

*"In recent years, as an open source and free model, SWMM has been coupled with other models, such as BreZo (Burns et al., 2015) and LISFLOOD-FP (Wu et al., 2017), to simulate urban inundation, which means that the coupled models based on SWMM is needed in future research."*

3. Page 4, line 8: I think the formulation 'useless nodes and pipelines' is chosen not so well. What is a useless node or pipeline? In reality each pipe was built for a purpose, so in principle no part of the pipe network is useless.

Re: Indeed, these 'useless nodes and pipelines' are independent of model building and just part of data cleaning. Our expression in data processing is not scientific enough and this sentence has been modified.

*Change in manuscript: This sentence has been modified on Page4 line 13.*

*"and delete the nodes and pipelines that independent of this model."*

4. I would suggest to merge Sections 2.3 and 2.4, as both just describe the model that was applied. The algorithm how water is exchanged between the surface and the pipe network is still not clear to me. Maybe it is not so crucial for the paper, as its focus is not any more on the model.

Re: We merged Sections 2.3 and 2.4.

After reading the manual of IFMS Urban, we briefly introduce the algorithm of couplying here. Like other coupled models, the first step is to calculate the exchanged water, and then substitute the results into the respective model to calculate and update to the next step. The formula of exchanged water is shown below:

$$Q = M\left(H_{node} - H_{surface}\right)W_{crest}\sqrt{2g\left|H_{node} - H_{surface}\right|}\frac{\left|H_{node} - H_{surface}\right|}{\left|\max(H_{node}, H_{surface}) - H_g\right|}$$

$H_{surface}$ is ground head; $H_{node}$ is pipe head; M is flow coefficient; $H_g$ is elevation.

*Change in manuscript: Sections 2.3 and 2.4 have been merged on Page 4, line 28-29.*

5. Section 2.5: I think it would be good to give some explanation why these two measures were chosen. Also, it is not very clear to me what is meant by the percentages. It is clear that it means that a certain percentage of all possible GR measures (for example) is considered. But at which locations were the roofs chosen that lead to a certain percentage of all possible measures? I am quite sure that it matters where they would be located (effective close to hot spots, less effective further away). Were they distributed equally over the domain?

Re: Green roof and permeable pavement are representative LID practices for urban inundation mitigation, and they have been applied on local area. Therefore we chose these two typical LID practices.

Many stduies set different levels of LID implementation to evaluate the effects of LID. For example, Hu et al. (2017) considered 50% and 75% as the implementation levels of permeable pavement, Zhang et al.(2016) set 50% and 100% as the implementation levels of three LID practices, Ahiablame and Shakya (2016) set four levels of implementation from 25%-100% of all LID practices. Similar scenarios can be found in many studies (Luan et al., 2017; Palla and Gnecco, 2015). Like most studies, the percentages in this paper also mean the levels of LID implementation. As for the locations of LID practices in this paper, we established two principles as follows: GR can be built on low density construction land, and PP can be built on low and high construction land and on some streets (Page 5 line 11), which can be found at Figure 1. And they distributed equally over their avaliable area like other studies.

[Figure]

**Figure 1: Location and land use map of the study area in the Guangming New District of Shenzhen, China.**

*Change in manuscript: No change.*

6. Page 6, line 12: How can one conclude that the cause of the high level is due to severe waterlogging? It might be the cause in reality, but the model will only reproduce causes that are in the model. How were waterlogging conditions implemented in the model?

Re: Originally, we aimed to express that: through the comparison, we found most inundated areas were not easy to be eliminated at the high level. We have modified the sentence.

*Change in manuscript: The sentence has been modified on Page 6 line 19.*

7. Section 4.1: I find it still a problem that the influence of storage is not discussed at all. The effects discussed in the manuscript are certainly caused by the storage assigned to the LID measures. Green roofs have a certain volume. If it is full, no more water can be held back. With more volume, more water could be held back. Permeable pavement gives access to the soil, which is a very large storage. But there is an infiltration rate, so that the storage can only be filled with a certain speed. I think a lot of the effects can be explained with this aspect and it should be discussed

Re: In this study, we compared the effects of green roof and permeable pavement. Similar comparisons were carried out in previous studies, such as permeable pavement, green roof and rain barrel (Zhang et al., 2016), porous pavement, rain barrel and rain garden (Ahiablame and Shakya, 2016), permeable pavement and rainwater harvesting (Hu et al., 2017), permeable pavement, Concave greenbelt, Bio-retention, et al. (Luan et al., 2017). Through the comparisons we can find that the characteristics of LID, implementation area, rainfall intensity and other factors are not same in these research. Indeed, not only the thickness of the storage layer, but also the porosity of soil layer, void ratio of pavement layer and many other parameters that can influence the effects of

LID practices. As requested, we did a sensitivity analysis to better identify the effects of parameters, and similar analysis can be found at Qin (2013).

*Change in manuscript: We did a lot changes on Page 7 line 23-Page 8 line 3.*

*"To better identify the effects of parameters, we did a sensitivity analysis carried out by assuming a 50% increase in some parameters under scenario 7-8, and the results showed that the inundation depth has great sensitivities to some parameters (Table 6). Under the permeable pavement scenario, the inundation decreases 15 %, 16 % and 18 % with thickness of pavement layer, thickness and void ratio of storage layer, respectively. Under the rain roof scenario, the inundation decreases 17 % and 19 % with thickness and porosity of soil layer, respectively. The results indicate that LID parameters might influence the effectiveness on inundation mitigation.*

*Indeed, except the LID parameters, there are some other factors, such as implementation area, spatial pattern, rainfall intensity and rainfall frequency that will influence the effectiveness, and these are the reasons why PP cannot always perform better and showed varying effectiveness in different studies (Ahiablame and Shakya, 2016; Hu et al., 2017; Qin et al., 2013; Zhang et al., 2016). However, under certain scenarios of this study, PP may be a good choice for local governments because of its effectiveness for stormwater management and its potential use for reconstruction in built-up areas."*

8. The finding that 25 percent of both measures are more efficient than 100 percent of one measure can also be explained with the storages assigned to the LID measures, I assume. A certain storage volume is needed to hold a certain flood volume back. So if measures are taken that provide this volume, the measure is fully efficient. If more is provided, it does not add to further mitigation. In the paper one gets the impression that the fact that less percentage of several measures is more efficient than more percentage of one measure is an unexplainable fact that is here 'found'. In general, the results should be discussed more to get to general conclusions

9. Section 4.3: The cost-effectiveness indication is newly introduced and was not part of the first manuscript. I am not convinced by this indicator. It has a unit that depends on the analyzed quantity and the range would go to infinity if no measures are taken. It is a very non-linear function of the percentage of the measures. This makes it difficult to interpret. If it should be used, it needs to be described very clearly what a certain number of the CEI means. The index comes out of the blue and it is nowhere discussed why this is a good index for the effectiveness of a LID measure.

Re: In the construction of Sponge City, people paid more attention to the effectiveness, while ignored the cost of LID. If we pursued the best effectiveness, we would not support such huge cost. Therefore, we want to find wheather there is a best efficiency scenario considering both the effectiveness and the cost. The cost-effectiveness index has been applied on Wu et al. (2017), although CEI might not be the best index for cost-effectiveness analysis, considering the available data and reasonable assumption, it was acceptable in this paper.

After adding two scenarios, 5 % GR + 5 % PP and 10 % GR + 10 % PP, we further verified the conclusion that: wider implementation of LID practices may not lead to higher efficiency, and there exist a highest efficiency scenario. Therefore, we should not only consider the effectiveness but also the cost during the construction of Sponge City.

Here we will discuss the causes of the phenomenon. In urban watershed, the runoff might not only come from sub-catchments around the inundation areas, but also come from other sub-catchments through the roads and pipe networks. And in this study, there are still some areas that can not implement LID practices. Therefore, the runoff from these areas can not be controlled by LID practices and directlly influenced the effectiveness of inundation mitigation. For example, the reduction rates of maximum inundation depth are 7, 16, 22, 26 and 29 % from scenario 2-6, and we can clearly see that the efficiency of LID practices has reduced continuously, especially from scenario 4-6 (the levels of implementation were from 50 % to 100%, but the reduction rates only increased 7 %). Therefore, wider implementation of LID practices may not lead to higher efficiency, and there exist a highest efficiency scenario in this study.

The phenomenon is common. In urban watershed, we could not transform all the roofs and roads to LID practices, and there are still some impervious covers that could influence the inundation that LID practices can not control. Therefore, we should recognize the insufficients of LID practices in the practical applications, and consider combine other measures such as restoring river systems, establishing urban wetlands, and improving urban drainage infrastructure to further promote the effectiveness on inundation mitigation. Besides, properly implementing construction intensity of LID practices to achieve optimal efficiency in urban watershed will be very important for the construction of Sponge City. From this point of view, the conclusions drawn from the study are still very attractive and have certain scientific value.

*Change in manuscript: We did a great changes on Section 4.3 (Page 8 line 26-Page 9 line 10).*

*"And we can clearly find that the reduction rates of maximum inundation depth are 7, 16, 22, 26 and 29 % from scenario 2-6 and the CEI has reduced continuously, especially from scenario 4-6. This indicates that wider implementation of LID practices may not lead to higher efficiency.*

*One of the causes behind the phenomenon is that LID practices can not control all the runoff of the watershed. Indeed, the runoff might not only come from sub-catchments around the inundation areas, but also come from other sub-catchments through the roads and pipe networks. And in this study, there are still some areas that can not implement LID practices. Therefore, the runoff from these areas can not be controlled by LID practices and directlly influenced the effectiveness of inundation mitigation.*

*The phenomenon is common. In urban watershed, we could not transform all the roofs and roads to LID practices, and there are still some impervious covers that could influence the inundation that LID practices can not control. Therefore, we should recognize the insufficients of LID practices, and consider combine other measures such as restoring river systems, establishing urban wetlands, and improving urban drainage infrastructure to further promote the effectiveness on inundation mitigation. Besides, properly implementing construction intensity of LID practices to achieve optimal efficiency in urban watershed will be very important for the construction of Sponge City."*

10. Page 9, lines 4-5: I do not see how the conclusion can be made that the model can be used for different cities and countries. In principle, of course any model could be used, but if the meaning is just that, the sentence is trivial. I would delete this sentence.

Re: Amended as requested.

*Change in manuscript: We have deleted this sentence on Page 9 line 30.*

**A list of modifications related to comments**

Please notice that page and line numbers are those of the revised version. And because of the addition of sensitivity analysis, the Table 7 is the old Table 6.

[revised manuscript text omitted]